# T-SNE EXAGGERATES CLUSTERS, PROVABLY

**Noah Bergam[†], Szymon Snoeck[*], & Nakul Verma[†]**
[†]Computer Science Department; [*]Applied Mathematics Department
Columbia University
{noah.bergam,sgs2179}@columbia.edu, verma@cs.columbia.edu

## ABSTRACT

Central to the widespread use of t-distributed stochastic neighbor embedding (t-SNE) is the conviction that it produces visualizations whose structure roughly matches that of the input. To the contrary, we prove that (1) the strength of the input clustering, and (2) the extremity of outlier points, *cannot* always be inferred from the t-SNE output. We demonstrate the prevalence of these failure modes in practice as well.

## 1 INTRODUCTION

t-SNE (Van der Maaten & Hinton, 2008) and related data visualization methods have become staples in modern exploratory data analysis. They just seem to work: practitioners find that these techniques reliably tease out interesting cluster structures in datasets. Consequently they are now used ubiquitously in a wide array of fields, ranging from single-cell genomics to language model interpretability (Kobak & Berens, 2019; Petukhova et al., 2025). The practical success of these techniques has naturally piqued interest in the theoretical community as well.

Existing analysis of t-SNE has established that, given high-dimensional data with spherical, well-separated cluster structure, t-SNE outputs a visualization which preserves that cluster structure (Arora et al., 2018; Linderman & Steinerberger, 2019). In other words, t-SNE is provably good at generating *true positives* in its visualization of clusters.

Curiously, a theoretical investigation of t-SNE's potential to generate *false positives* (clustered plots despite un-clustered input) and *false negatives* (un-clustered plots despite clustered input) has remained elusive. One should note that this is not a purely academic curiosity, since the interpretation of t-SNE outputs have important consequences downstream in the sciences, influencing hypothesis generation, experimental design, and scientific conclusions.

As an illustration of t-SNE's potential to mislead, consider the visualizations produced by three example inputs in Figure 1. In the left column, a t-SNE plot depicts two salient clusters, despite the pairwise distances in the input being nearly uniform. In the middle column, a t-SNE plot depicts an unstructured blob, whereas its corresponding input is easily partitioned into two salient clusters (with the exception of a single adversarially placed point). In the right column, a t-SNE plot shows two well-separated clusters, completely misrepresenting the outlying third of the input dataset.

Our work formalizes these and other limitations of t-SNE in terms of faithfully depicting distance-based information in the input. Our theoretical analysis, suffused with experiments, shows that one should take t-SNE visualizations with a grain of salt. Our contributions are as follows.

(i) **Misrepresentation of cluster salience:** We prove that *any* well-clustered t-SNE visualization can be produced identically by both a strongly-separated, as well as arbitrarily weakly-separated clustered datasets, see Theorem 3 and Corollary 4. Moreover, we prove that even a slight distance perturbation of inputs can lead to vastly distinct visualizations, see Theorem 5. We identify the property of t-SNE that explains these peculiar behaviors, and use this understanding to design an adversarial attack that disrupts cluster structure in the output by moving as little as one point, see Figure 4.

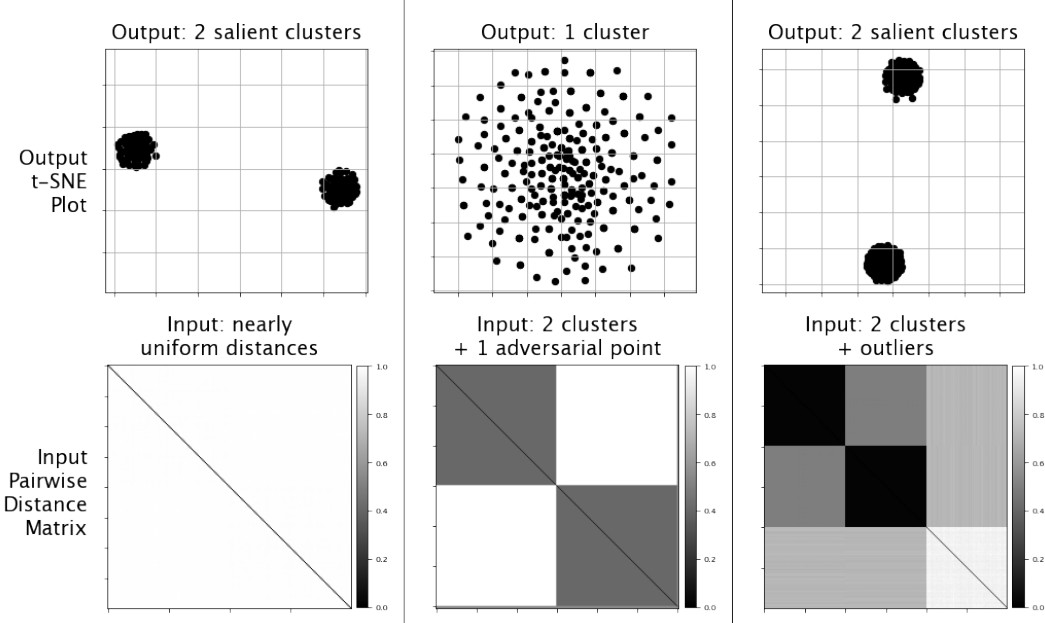

Figure 1: Some examples of how t-SNE may misrepresent cluster information in the input dataset.

(ii) **Misrepresentation of outliers**: We prove that, regardless of input, the resulting t-SNE output is incapable of depicting extreme outliers, in the sense of depicting one point as substantially far away from all the others, see Theorem 9. In practice, on both synthetic and real datasets, we observe a more concerning phenomenon that faraway outliers are often subsumed into the cluster structure of the bulk of points, see Figures 5 and 6.

While there has been some work investigating the shortcomings of t-SNE in various practical settings (see Section 2.2 for a detailed discussion of the relevant literature), to the best of our knowledge this is the first work which theoretically analyzes some of the key limitations of t-SNE.

## 2 RELATED WORK

Confidence in the data visualizations produced by t-SNE and related methods is a contentious subject in data science (Marx, 2024; de Bodt et al., 2025). Some argue that these methods have merit in terms of preserving cluster structure and therefore aid in exploratory data analysis, while others warn us about the distortions introduced by these methods.

### 2.1 PERFORMANCE GUARANTEES AND ANALYSIS OF T-SNE

Shaham & Steinerberger (2017) were among the first to provide a guarantee on the visualization produced by optimal SNE embeddings of well-clustered data. Works by Linderman & Steinerberger (2019) and Arora et al. (2018) refined and extended this analysis, showing that t-SNE outputs produced using gradient descent yield well-clustered visualizations so long as the input is sufficiently well-clustered. The latter work established this guarantee in considerable generality, including cases where the input is sampled from a mixture of well-separated log-concave distributions.

Along with these algorithmic performance guarantee results, there is a line of work that seeks to establish a more fundamental understanding of t-SNE as an optimization problem. Cai & Ma (2022), for instance, characterized the distinct phases of gradient-based optimization of t-SNE, and proved an asymptotic equivalence between the early exaggeration phase of t-SNE and spectral clustering. Auffinger & Fletcher (2023) proved a consistency result for a continuous analogue of t-SNE, viewing the optimization problem as producing a map between distributions rather than just point sets. Jeong & Wu (2024) and Weinkove (2024) studied the gradient flow of t-SNE. The former showed mild

assumptions under which optima exist, and the latter showed that, even in cases where the gradient flow diverges the relative interpoint distances stabilize in the limit.

## 2.2 WEAKNESSES AND CRITICISMS

Bunte et al. (2012) were among the first to investigate the potential shortcomings of using KL-divergence in a t-SNE visualization and proposed a generalization to other divergences that may be better suited for specific datasets and user needs. Building upon the precision-recall framework of Venna et al. (2010), Im et al. (2018) extended this result and explored specific intrinsic structures within data that may be less suited for t-SNE. They concluded that while t-SNE is more attuned to reveal intrinsic cluster structure, it usually fails to reveal intrinsic manifold structure.

In terms of analyzing cluster structure specifically, Yang et al. (2021) provided empirical evidence that t-SNE visualizations are prone to *false negatives*. They presented a selection of well-clustered real-world datasets which t-SNE embeddings, even with reasonable parameter-tuning, do not seem to represent faithfully. They also showed that these practical datasets do not abide by the theoretical cluster separation conditions that are required by Arora et al. (2018)'s analysis. Chari & Pachter (2023) argued that t-SNE and UMAP are unreliable tools for exploratory data analysis. Taking single-cell genomic data as an important real-world example, they provided systematic empirical evidence that these embeddings suffer high distortion, and often misrepresent neighborhood and cluster structure. Curiously, to the best of our knowledge, there is no systematic theoretical study investigating the failure modes of t-SNE.

More recently, Snoeck et al. (2026) provided theoretical evidence that, not just t-SNE, but any embedding technique that attempts to visualize data in constant dimensions is bound to misrepresent neighborhood structure in almost all inputs. This result makes a general assertion that misrepresentation is geometrically inevitable; the current work investigates how such misrepresentations manifest precisely in t-SNE on a variety of inputs of interest.

## 3 PRELIMINARIES

Given an input dataset $X = \{x_1, \ldots, x_n\} \subset \mathbb{R}^D$, the goal of t-SNE is to come up with an embedding $Y = \{y_1, \ldots, y_n\} \subset \mathbb{R}^d$ (where $d \ll D$, typically $d = 2$)[1] that approximately maintains the neighborhood structure in $X$. t-SNE accomplishes this by assigning affinities to input data points which encode how likely an input point is to be a neighbor to a given point. The goal then is to find a configuration of the embedded points $Y$ that induces a similar neighborhood affinity. Specifically, let $P = P(X) \in \mathbb{R}_{\geq 0}^{n \times n}$ and $Q = Q(Y) \in \mathbb{R}_{\geq 0}^{n \times n}$ be the input and embedded *affinity matrices* describing the pairwise neighborhood similarities in the input and output, respectively. t-SNE constructs $P$ by first computing neighborhood affinities for each point $i$ defined as (for any $i \neq j$)

$$P_{j|i}(X; \sigma_i) := \frac{\exp(-\|x_j - x_i\|^2/(2\sigma_i^2))}{\sum_{k \neq i} \exp(-\|x_k - x_i\|^2/(2\sigma_i^2))} \qquad P_{i|i}(X; \sigma_i) := 0 \qquad (1)$$

where $\sigma_i \geq 0$ encodes the (point-dependent) neighborhood scaling[2]. When $X$ and $\sigma_i^*$ are clear from context, we will often drop it from the notation. It is worth noting that $P_{\cdot|i}$ is a valid probability distribution over $[n]$. The matrix $P$ is then constructed based on a crucial parameter called the perplexity (denoted as $\rho$ and taking values in $[1, n-1]$), as follows:

(1) For each $i \in [n]$, select the (unique, see Lemma 14) neighborhood scale $\sigma_i^*$ that minimizes the gap between the entropy of $P_{\cdot|i}(X; \sigma_i^*)$ and $\log_2 \rho$.

(2) Define $P = [P_{ij}]_{i,j \in [n]}$ where $P_{ij} := \frac{1}{2n}(P_{i|j}(\sigma_j^*) + P_{j|i}(\sigma_i^*))$.

To avoid the so-called *crowding problem* (see Van der Maaten & Hinton (2008) for details), the output affinity matrix $Q$ is computed based on a t-distribution. Specifically, for $i \neq j$

$$Q_{ij}(Y) := \frac{(1 + \|y_i - y_j\|^2)^{-1}}{\sum_{k,l; k \neq l}(1 + \|y_k - y_l\|^2)^{-1}} \qquad Q_{ii}(Y) := 0. \qquad (2)$$

---

[1]With a slight abuse of subset notation, duplicates are allowed in $X$ and $Y$.

[2]If $\sigma_i = 0$, define $P_{j|i}(X; 0) := \lim_{\sigma_i \to 0} P_{j|i}(X; \sigma_i)$.

As indicated before, the objective then is to minimize the gap between the input and output affinities $P$ and $Q$. This is accomplished by minimizing relative entropy (KL-divergence) between the $P$ and $Q$ affinities (viewed as probability distributions).

$$\text{minimize}_Y \ \mathcal{L}_X(Y) := \text{KL}(P(X)\|Q(Y)) = \sum_{\substack{i,j \\ i \neq j}} P_{ij}(X) \log \left( \frac{P_{ij}(X)}{Q_{ij}(Y)} \right).$$

This highly non-convex objective is usually optimized by initializing at a good starting point via an *early exaggeration phase*, followed by performing standard gradient descent methods and returning an embedding $Y$ that corresponds to a local minimum of the objective. Our central task is to study the nature of the these (local minimum) embeddings returned by t-SNE and their relation to the space of input datasets.

**Definition 1.** *For an $n$-point dataset[3] $X \subset \mathbb{R}^{n-1}$ and perplexity parameter $\rho \in [1, n-1]$, define*

$$\text{t-SNE}_\rho(X) := \{Y \subset \mathbb{R}^d : \nabla_Y \mathcal{L}_X(Y) = 0\}$$

*as the set of outputs $Y \subset \mathbb{R}^d$ that are stationary to the t-SNE objective on a given input $X$. Furthermore, for a set of $n$-point datasets $\mathcal{X}_n$, we define:*

$$\text{t-SNE}_\rho(\mathcal{X}_n) := \bigcup_{X \in \mathcal{X}_n} \text{t-SNE}_\rho(X).$$

*If $\mathcal{X}_n$ is the set of* all *$n$-point datasets, we denote $\text{t-SNE}_\rho(\mathcal{X}_n)$ as $\text{Im}(\text{t-SNE}_{\rho,\text{n}})$ to indicate the entire image of the t-SNE map.*

All the supporting proofs for our formal statements can be found in the Appendix, and the code related to our experimental demonstrations is available on Github at `https://github.com/njbergam/tsne-exaggerates-clusters`.

## 4 MISREPRESENTATION OF CLUSTER SALIENCE

Previous works by Linderman & Steinerberger (2019) and Arora et al. (2018) have identified that clustered inputs induce clustered t-SNE visualizations in the sense that sufficiently well-separated Gaussian-shaped clusters in the input must produce corresponding well-separated clusters in the output visualization. A key question for practitioners left unanswered by these analyses is: when does a clustered output imply a clustered input? More generally, what information can be deduced about the input given a visualization? We begin to answer this question by proving that the strength of cluster separation in the input cannot be reliably inferred from the low-dimensional visualization.

To quantify the strength of cluster separation in a dataset, we employ well-known distance-based cluster indices such as the average silhouette score (Rousseeuw, 1987), the Calinski-Harabasz index (Caliński & Harabasz, 1974), and the Dunn index (Dunn, 1974). For sake of readability, we focus on presenting our results with respect to the average silhouette score. Our results hold identically for the other indices as well (see Appendices A.4 and A.5).

**Definition 2.** *Given a partition $C_1 \sqcup C_2 \sqcup \cdots \sqcup C_k = [n]$ of $n$ points $\{x_1, \ldots, x_n\} = X$ into $k \geq 2$ parts, the **silhouette score** of a point $x_i$, denoted $\mathcal{S}(i)$, is the normalized difference between the average within- and the closest across-cluster distances from $x_i$:*

$$\mathcal{S}(i) := \frac{b(i) - a(i)}{\max\{b(i), a(i)\}} \qquad a(i) := \sum_{j \in C^{(i)}} \frac{\|x_i - x_j\|}{|C^{(i)}| - 1} \qquad b(i) := \min_{\substack{m \in [k] \\ C_m \neq C^{(i)}}} \sum_{j \in C_m} \frac{\|x_i - x_j\|}{|C_m|},$$

*where $C^{(i)}$ is the cluster to which $i$ belongs. Note that if $|C^{(i)}| = 1$, then $\mathcal{S}(i)$ is defined to be zero. The **average silhouette score** then is simply the average across all points in $X$:*

$$\bar{\mathcal{S}}(X; C_{m \in [k]}) := \frac{1}{n} \sum_{i \in [n]} \mathcal{S}(i).$$

Observe that the (average) silhouette score ranges from $-1$ to $1$ with higher scores reflecting a large separation between clusters relative to cluster diameter. A score of zero reflects minimal separation between clusters, while negative values reflect cluster overlaps.

---

[3]Without loss of generality, we shall assume that the input dimension $D = n - 1$.

## 4.1 Different Inputs, Same Output

Defining the strength of a clustering with respect to the silhouette score, we show that any stationary t-SNE output (including outputs with arbitrarily well-separated clusters) can be produced by an input with minimal distance separation between the clusters:

**Theorem 3.** *Fix any $n > k > 1$, and $n$-point dataset $X \subset \mathbb{R}^{n-1}$ with partition $C_1 \sqcup \cdots \sqcup C_k = [n]$ such that $|C_{m \in [k]}| > 1$ and $\bar{\mathcal{S}}(X; C_{m \in [k]})$ is well defined. For all $0 < \epsilon \leq 1$, there exists $n$-point dataset $X_\epsilon \subset \mathbb{R}^{n-1}$ such that*

$$\bar{\mathcal{S}}(X_\epsilon; C_{m \in [k]}) = \epsilon \cdot \bar{\mathcal{S}}(X; C_{m \in [k]}),$$

*yet, for any $\rho \in [1, n-1]$,*

$$\text{t-SNE}_\rho(X) = \text{t-SNE}_\rho(X_\epsilon).$$

It is important to understand the implications of this result. For any high-dimensional dataset $X$ (that contains well-separated clusters), we can always find an impostor dataset $X_\epsilon$ with minimal cluster separation such that *all* t-SNE (local as well as global) stationary points of $X$ and $X_\epsilon$ match perfectly! In other words it is *impossible* to distinguish between $X$ and $X_\epsilon$ based on the low-dimensional t-SNE visualization.

As a consequence, the same visualization of salient clusters can be produced by a sequence of impostor datasets containing clusters ranging from well-separated to minimally separated.

**Corollary 4.** *For all $n \geq 4$ even, and partition $C_1 \sqcup C_2 = [n]$ such that $|C_1| = |C_2| = \frac{n}{2}$. There exist a sequence of $n$-point datasets in $\mathbb{R}^{n-1}$, $\{X_\epsilon\}_{0 < \epsilon \leq 1}$, with*

$$\bar{\mathcal{S}}(X_\epsilon; C_1, C_2) = \epsilon$$

*such that for any $\rho \in [1, n-1]$, we have $Y \in \bigcap_{0 < \epsilon \leq 1} \text{t-SNE}_\rho(X_\epsilon)$ with*

$$\bar{\mathcal{S}}(Y; C_1, C_2) = 1.$$

The above shows that $Y$, a perfectly clustered visualization according to silhouette score, is a local (and global, see the proof in Appendix) minimizer for any member of a set of inputs of *arbitrary* silhouette score. Thus, even from a visualization which is *perfectly* clustered, the strength of the input's cluster structure cannot be inferred.

Note that the existence of an impostor $X_\epsilon$ is not just theoretical; it can be constructed practically as well (see Appendix A.7 for an explicit construction). Hence this phenomenon can be demonstrated in real-world scenarios, see Figure 2. In this case, we select a preprocessed version of the well-known PBMC3k single-cell genomics dataset (2638 points, 50 dimensions; 10x Genomics (2019)) as $X$. We show that there is an "impostor dataset" $X_\epsilon$ that is essentially indistinguishable from the real dataset in terms of its 2D t-SNE visualization, yet has a much weaker cluster separation than the original dataset. The difference between impostor and original can be quantified by silhouette score and visualized in terms of the interpoint distance matrix and dendrogram.

It is worth emphasizing that while the distance information in the original and impostor dataset is dramatically different, the *ordinal* relationship between neighbors is identical between the datasets, as demonstrated by the corresponding structure in the dendrograms.

## 4.2 Different Outputs, Similar Inputs

The previous section established that inputs with qualitatively distinct metric structure can yield the exact same t-SNE output. We continue with a complementary result: that near-identical inputs can yield very distinct outputs.

### 4.2.1 Instability under small distance perturbations

**Theorem 5.** *Fix any $n \geq 2$ and $\rho \in [1, n-1]$. For all $\epsilon > 0$ and all $Y, Y' \in \text{Im}(\text{t-SNE}_{\rho,n})$, there exists $n$-point datasets $X = \{x_1, \ldots, x_n\}$ and $X' = \{x'_1, \ldots, x'_n\} \subset \mathbb{R}^{n-1}$ such that $\forall i \neq j$*

$$1 - \epsilon \leq \frac{\|x_i - x_j\|^2}{\|x'_i - x'_j\|^2} \leq 1 + \epsilon,$$

*yet $Y \in \text{t-SNE}_\rho(X)$ and $Y' \in \text{t-SNE}_\rho(X')$.*

Figure 2: Visualizations of single-cell data (top row) versus an impostor dataset with arbitrarily small cluster separation (bottom row). Based on the 2D t-SNE visualization (left column), it is difficult to distinguish which dataset (real or impostor) may have produced the plot. Plotting the input interpoint distance matrices (middle column) suggests that the clusters in the impostor dataset are significantly less separated than in the original dataset. The corresponding dendrograms (right column, produced using the Ward's method) further elucidate the relative strength of pairwise distances. It is worth emphasizing that the impostor dataset retains the relative rank ordering of the pairwise distances (and therefore the ordering of nearest neighbors), and only distorts the distances to make the differences much finer. Note that the color coding in all of the scatterplots corresponds to a stable cut in the hierarchical clustering given by the dendrogram of the original dataset, and the bottom left labels correspond to silhouette scores with respect to that clustering.

Thus arbitrarily minor interpoint distance perturbations of the input dataset can develop into massive changes in the visualization. Figure 3 demonstrates this phenomenon quite clearly. We start with a dataset $X$ that is a regular unit simplex (all pairwise distances are unit length). By systematically perturbing the input $X$ ever so slightly ($\epsilon \leq 0.01$), t-SNE produces strikingly different outputs.

The key observation behind our main Theorems 3 and 5 is the simple yet counter-intuitive fact that t-SNE is not only invariant under multiplicative scaling of the input distances, but also *additive* shifts thereof – a property also investigated independently by Lee & Verleysen (2011; 2014). Specifically given a dataset $X = \{x_1, \ldots, x_n\}$, for any dataset $X' = \{x'_1, \ldots, x'_n\}$ and $C \in \mathbb{R}$ such that, $\|x'_i - x'_j\|^2 = \|x_i - x_j\|^2 + C \geq 0$ for $i \neq j$, we have t-SNE$_\rho(X) = $ t-SNE$_\rho(X')$ (see Lemma 17 for a formal statement, and Lee & Verleysen (2011), Section 4). As a consequence, for any input dataset, we can simply pump up the interpoint distances and construct an impostor dataset which has the same visualization profile but is arbitrarily close to a regular simplex (and hence is arbitrarily unclustered)[4]. This observation also leads to the following seemingly bizarre fact.

**Lemma 6.** *Fix any $n \geq 2$ and $\rho \in [1, n-1]$. For any $\epsilon > 0$, define the set of $\epsilon$-perturbations of a unit simplex as $\Delta_\epsilon := \{X = \{x_1, \ldots, x_n\} \subset \mathbb{R}^{n-1} : \forall i \neq j, \|x_i - x_j\|^2 \in [1 - \epsilon, 1 + \epsilon]\}$. Then, for all $\epsilon > 0$*

$$\mathsf{Im}\big(\text{t-SNE}_{\rho,\mathrm{n}}\big) = \text{t-SNE}_\rho(\Delta_\epsilon).$$

---

[4]See Algorithm 1 for a formalization of this process.

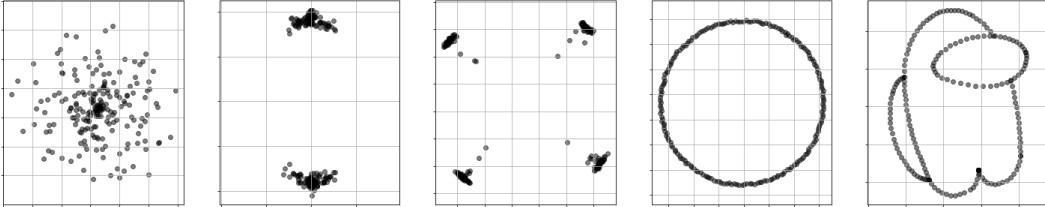

Figure 3: Various different 2D t-SNE visualizations produced by adversarial perturbations of a 200-point unit regular simplex. Each pair of perturbations satisfies the conditions of Theorem 5 for $\epsilon = 0.01$.

In other words there is a set of datasets $\Delta_\epsilon$ arbitrarily close to a regular unit simplex that generates *all* possible stationary t-SNE outputs! This result indicates that t-SNE outputs are highly unstable on near-simplex inputs (cf. Figure 3), which has real-world consequences since many high-dimensional datasets fall into this regime (Beyer et al., 1999; Aggarwal et al., 2001) due to the concentration of measure phenomenon (Ledoux, 2001).

### 4.2.2 INSTABILITY UNDER INSERTION OF A SINGLE POINT

The previous lemma tells us that on intrinsically high-dimensional data, small perturbations of all the interpoint distances can have outsize effects on the t-SNE output. We observe that such datasets are susceptible to a much simpler adversarial attack: namely, the insertion of a *single* data point.

Consider a dataset $X$ sampled from a mixture of two high-dimensional Gaussians. t-SNE, as expected, reveals the two underlying clusters (see Figure 4, first panel). However, we can add just a single "poison point" to $X$ and destroy the clustered visualization (see Figure 4, second panel; see also Figure 10). This failure mode of t-SNE is also observed on a real high-dimensional datasets (see Figures 6 and 15, left vs. center).

How can a single point do so much damage? The answer lies in how t-SNE registers neighborhood information and its potential consequences in high dimensions. In the case of Figure 4, the poison point is placed at the mean of the dataset (this is captured clearly by the PCA plot). Due to the high dimensionality of the configuration (indeed, this mixture of two Gaussians is approximately a regular simplex) the poison point is the nearest neighbor to most of the points in the dataset. This significantly changes the input affinity matrix, downweighting the intra-cluster affinities and upweighting affinity for the poison point, resulting in a visualization which keeps points in tight proximity to their nearest neighbor (namely, the poison point).

This disproportionate amount of affinity towards the poison point is modulated by the perplexity value. As $\rho$ decreases, the proportion of affinity placed on the poison point increases. In the limiting case, when $\rho = 1$, *all* affinity is assigned to the poison point. We can formalize this case as follows:

**Theorem 7.** *Take $n \geq 4$ even. There exists two $n$-point datasets such that:*

- $X_0 \subset \mathbb{R}^{n-1}$ *has silhouette score* 0 *with respect to all partitions, and*

- $X_1 \subset \mathbb{R}^{n-1}$ *has silhouette score* $\frac{\sqrt{3}-1}{\sqrt{3}} \geq 0.42$ *with respect to some partition of points,*

*yet there exists $x_0 \in \mathbb{R}^{n-1}$ such that for $\rho = 1$, t-SNE$_\rho(X_0 \cup x_0) = $ t-SNE$_\rho(X_1 \cup x_0)$.*

Therefore the poison point attack can make it *impossible* to distinguish between a well-clustered and a completely un-clustered input based on their t-SNE loss landscapes (this is experimentally corroborated in Figure 11). Our practical demonstrations (e.g. Figures 4, 6, 10, 15) indicate that a similar level of degradation persists for typical settings of $\rho$ in the usage of t-SNE.

The poison point attack can be viewed as a manifestation of the *hubness phenomenon*, a signature of high-dimensional data where nearest-neighbor relationships can be highly asymmetric (Beyer et al., 1999; Aggarwal et al., 2001). While previous work has established that hubness reduction can improve t-SNE's performance on certain benchmarks (Amblard et al., 2022), we theoretically quantify and experimentally demonstrate the striking damage that a single hub-like point can do.

| t-SNE, original | t-SNE, + 1 poison point | PCA, original | PCA, + 1 poison point |
|---|---|---|---|
| 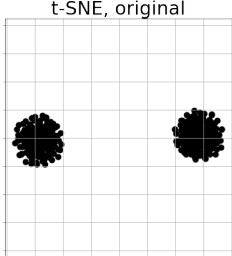 | 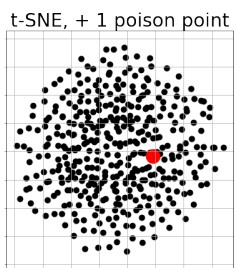 | 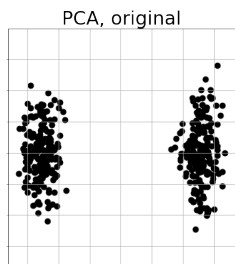 | 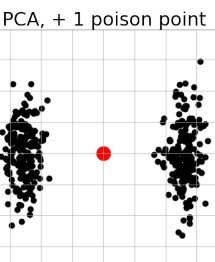 |

Figure 4: t-SNE versus PCA's radically different responses to the injection of a single "poison" point in the input dataset. The original dataset, visualized in panels 1 and 3, consists of 400 points sampled from a mixture of two Gaussians in $\mathbb{R}^{2000}$. The poison point is then placed at the mean of the previously sampled points; the resulting 401 point dataset is visualized in panels 2 and 4. Note that this behavior persists for larger datasets, see Appendix B.1.

## 5 MISREPRESENTATION OF OUTLIERS

Most analysis of t-SNE, including the previous section, is concerned with whether it faithfully depicts global structure, specifically cluster structure. In this section, we consider how t-SNE represents points that drastically deviate from the global structure: namely, outliers. It is natural to hope that data visualization methods can enable the identification of outliers. Unfortunately, we find that t-SNE may arbitrarily suppress the severity of outliers present in the input dataset.

This phenomenon has been observed empirically in prior work, though to our knowledge we are the first to formalize it (Schubert & Gertz, 2017). An intuitive explanation of t-SNE's response to outliers can be made based on the asymmetry of the input and output affinity matrices of t-SNE. Roughly speaking, the input affinity behaves like a normalized, symmetrized nearest neighbor graph, whereas the output affinity behaves more like a radius neighbors graph. This means the output affinity is optimized to represent the outlier point in close proximity with at least some points, even if it was extremely far from those points in the input.

To begin to formalize this observation, we provide a geometric definition of an outlier.

**Definition 8.** *Fix* $X \subset \mathbb{R}^D$ *and* $\alpha \in \mathbb{R}_{>0}$. *We say* $X$ *is an* $(\alpha, x_0)$-***outlier configuration*** *if there exists a hyperplane separating* $x_0$ *and* $X \setminus \{x_0\}$ *with margin width at least*

$$\alpha \cdot \max\{1, \text{diam}(X \setminus \{x_0\})\}.$$

*Define the* ***outlier number*** *of a dataset, denoted* $\alpha(X)$, *as the largest* $\alpha$ *for which there exists* $x_0 \in X$ *such that* $X$ *is an* $(\alpha, x_0)$-*outlier configuration.*

This definition can be generalized to accommodate more than one outlier. Note that $\alpha$ is defined with respect to a *thresholded* data diameter[5], i.e. $\max\{1, \text{diam}(X \setminus \{x_0\})\}$, to maintain a suitable notion of an outlier even in the extreme case of $\text{diam}(X \setminus \{x_0\}) = 0$.

Our main theorem establishes that any stationary t-SNE output, *regardless of its input*, is incapable of depicting extreme outliers.

**Theorem 9.** *Fix* $n > 2$ *and* $\rho \in [1, n-1]$. *Let* $Y \in \text{Im}(\text{t-SNE}_{\rho,n})$ *be any* *stationary t-SNE embedding. Then we have:*
$$\alpha(Y) \leq 3.266 + o_n(1).$$

The result is proven via analysis of the t-SNE gradient: we argue that if the outlier is too far away, its gradient is nonzero, thus violating stationarity. Key to this analysis is a comparison between the aggregate behavior of the outlier point's affinities in the input versus the output; in other words, the comparison between $\sum_{i=1}^{n} P_{i0}$ and $\sum_{i=1}^{n} Q_{i0}$. This is where the fundamental asymmetry of t-SNE comes in. While the latter is dependent on the position of the outlier point $y_0$, per Lemma 21, the former has a lower bound of $1/(2n)$ due to the normalization of the conditional affinity probabilities.

---

[5]The choice of a fixed threshold is arbitrary and does not meaningfully affect the results.

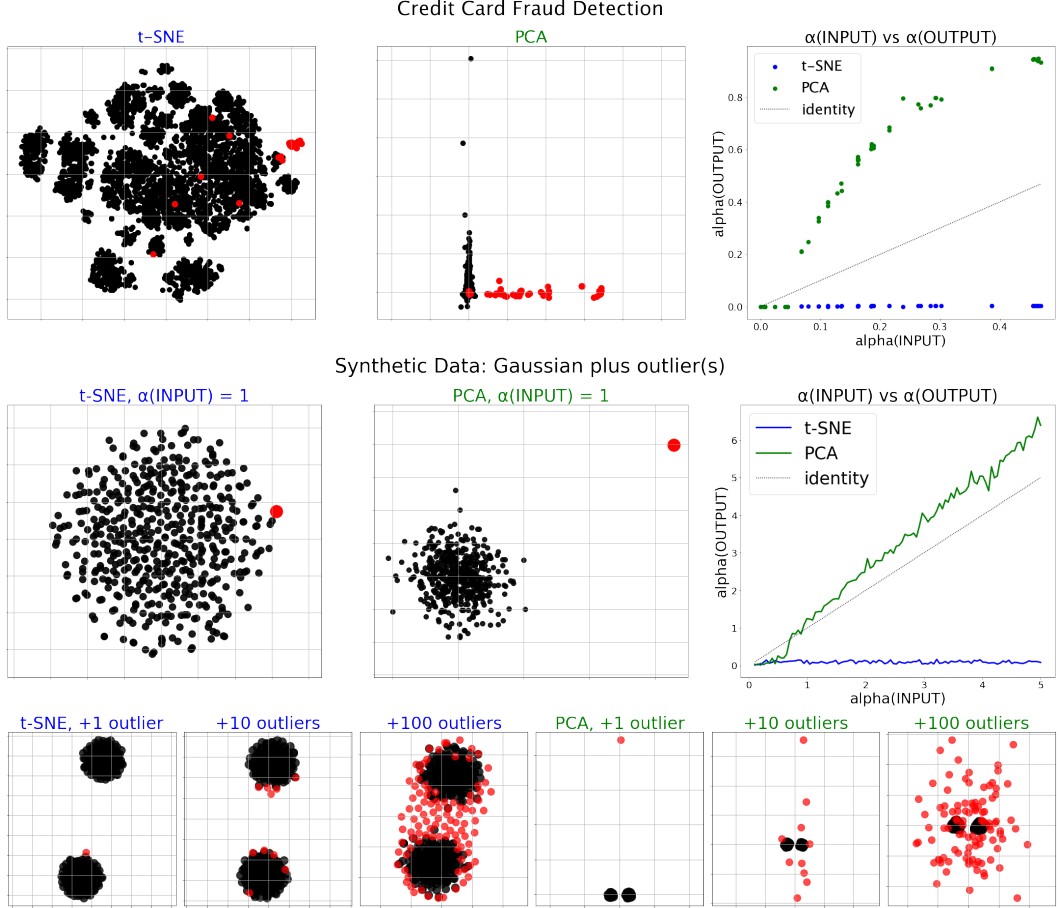

Figure 5: t-SNE's response to $\alpha$-outliers, compared with PCA. Top row: given data monitoring financial activity ($n = 5050$, $D = 30$) where one percent of users are committing fraud, PCA succeeds and t-SNE fails at representing the fraudulent users as outliers. Note that all of the fraudulent users register as ($\alpha > 0$)-outliers with respect to the regular users; in the top right we show how t-SNE and PCA represent those $\alpha$-values in their output. Middle row: a similar analysis on a synthetic dataset comprised of a Gaussian sample plus an outlier. Bottom row: mixture of two Gaussians plus 1, 10, and 100 outliers. t-SNE shows the outliers are essentially part of the cluster structure, while for PCA the outliers overtake the structure of the embedding.

The input-agnostic nature of this result is striking: even if the input is an extreme outlier configuration, no stationary t-SNE output can depict its extremity past roughly $\alpha \approx 3.2$. This behavior stands in stark contrast to that of principal component analysis (PCA), as shown in Figure 5 on both real and synthetic data models. PCA tends to roughly preserve the $\alpha$ outlier number, while t-SNE seldom depicts outliers past $\alpha > 0.2$ in practice, and sometimes even depicts them as within the convex hull of the rest of the points (yielding $\alpha = 0$). Furthermore, when faced with multiple outliers, (Figure 5, bottom) t-SNE gracefully accommodates them into the global structure of the bulk of the data, in contrast to PCA, which depicts the outliers as randomly scattered.

Our result suggests that t-SNE is the wrong tool to use in situations involving outlier detection. Consider, for instance, a dataset of financial transactions where the goal is to detect fraudulent users, studied by Pozzolo et al. (2015). In this dataset, only $0.172\%$ percent of the points (492 out of $284,807$) are fraudulent and by many standard statistical metrics register as outliers. Comparing the t-SNE and PCA plots on a random representative subset of this data (5050 points, of which 50 are fraudulent), we see that t-SNE mixes the frauds with the bulk of the points while PCA keeps them separated for the most part, see Figure 5, top row.

Figure 6: t-SNE's susceptibility to poison points in contrast with its muted response to outliers, on the BBC news article classification dataset. The bottom left label denotes silhouette score of the original points (without the injected points) with respect to the true labels (business, entertainment, politics, sport, tech).

Finally, note the distinction between t-SNE's muted response to outliers and its dramatic sensitivity to poison points. We illustrate this distinction on a dataset of BBC news articles which cluster into five topics (Greene & Cunningham, 2006), see Figure 6. Given RoBERTa (Liu et al., 2019) sentence embeddings of these articles ($n = 2225, D = 1024$), we find that injecting 220 poison points, i.e. roughly 10% (see Appendix B.1 for the explicit construction) can halve the silhouette score of the t-SNE embedding with respect to the ground-truth labelling, whereas injecting 1100, i.e. roughly 50%, $\alpha$-outliers for $\alpha \in (0.8, 1)$ does not degrade the cluster structure.

## 6 DISCUSSION

Our study of t-SNE has established in considerable generality that one cannot infer the *degree* of cluster salience or the extremity of outliers from a t-SNE plot, see Theorems 3, 5, and 9. The proofs and intuitions behind these statements guided us to surprising empirical observations that one cannot even infer the *existence* of clusters or outliers in some cases. In particular, the injection of a small subset of adversarially chosen points can largely mask the cluster structure, while sizable injections of outlier points are reliably masked within the cluster structure, see Figures 4, 5, 6, and 15.

We have identified two properties of t-SNE that give rise to these idiosyncratic behaviors: (1) additive invariance with respect to the squared interpoint distances, and (2) the asymmetry between the input and output affinity matrices. While we have uncovered significant failure modes that arise from these properties, we cannot completely rule out their utility. For instance, though additive invariance may lead to exaggerated clusters, prior work has indicated that it may be useful in filtering out high-dimensional noise that may be present in the input, see e.g. Figure 9, and Lee & Verleysen (2011; 2014); Karoui (2010); Karoui & Wu (2015); Landa & Cheng (2023).

t-SNE belongs to a wide selection of data visualization techniques that are yet to be understood fully (McInnes et al., 2018; Jacomy et al., 2014; Tang et al., 2016; Amid & Warmuth, 2019). Our preliminary experiments (see Appendices A.3 and B.2) suggest that the failure modes discussed in this paper may extend to other force-based dimension reduction techniques. Our hope is that this work inspires the reader to explore this fascinating landscape further and pursue the essential question: what can be provably deduced from a visualization?

## ACKNOWLEDGMENTS

The authors would like to thank the anonymous reviewer for their helpful feedback and suggesting improvements to the manuscript. N.B. was supported by the ONR grant N0001424SB001 and The Herbert and Florence Irving Institute for Cancer Dynamics during the writing of this paper.

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

## A    APPENDIX: MISREPRESENTATION OF CLUSTER SALIENCE

### A.1    EFFECTS OF PCA PRE-PROCESSING

It is common practice to pre-process a t-SNE input by reducing the dimension with PCA (Kobak & Berens, 2019). In light of the results of Section 4, it is natural to ask how this PCA pre-processing step plays with t-SNE's additive invariance. Figures 7 and 8 provide evidence that, even if the input is pre-processed with PCA, there exist a sequence of datasets with similar visualizations but vastly different input cluster strengths as measured via silhouette score and aspect ratio.

In both figures, the dataset at hand is 1000 points sampled from a mixture of two Gaussians in 100 dimensions. Algorithm 1 is applied at various strengths to vary the silhouette score and aspect ratio.

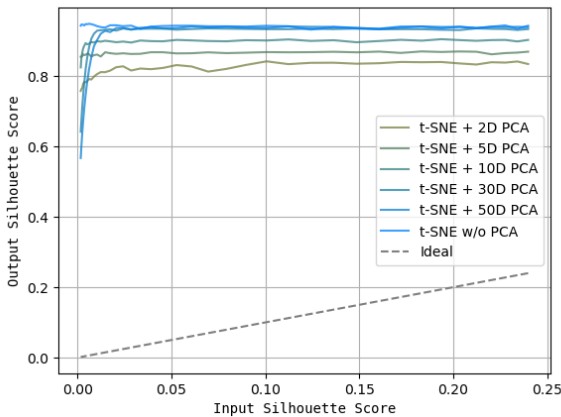

Figure 7: Effect of PCA Pre-processing on Silhouette Score: input versus t-SNE output silhouette score using various levels of PCA pre-processing.

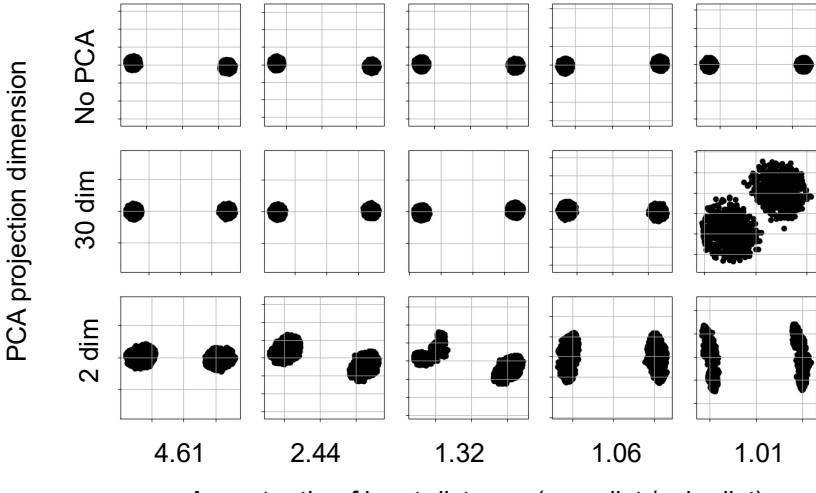

Figure 8: Effect of PCA pre-processing on visualization. Note that without PCA, t-SNE is perfectly additively invariant, whereas this is not the case with PCA pre-processing.

## A.2 Additional Experiments

In Figure 9, we plot a sample from a mixture of two Gaussians in 250, 500, 1000, 2000, and 4000 dimensions. Notice that as the dimension of the Gaussian increases, the interpoint distance matrix of the input points (bottom) approaches a simplex but the t-SNE corresponding visualization (top) remains qualitatively unchanged.

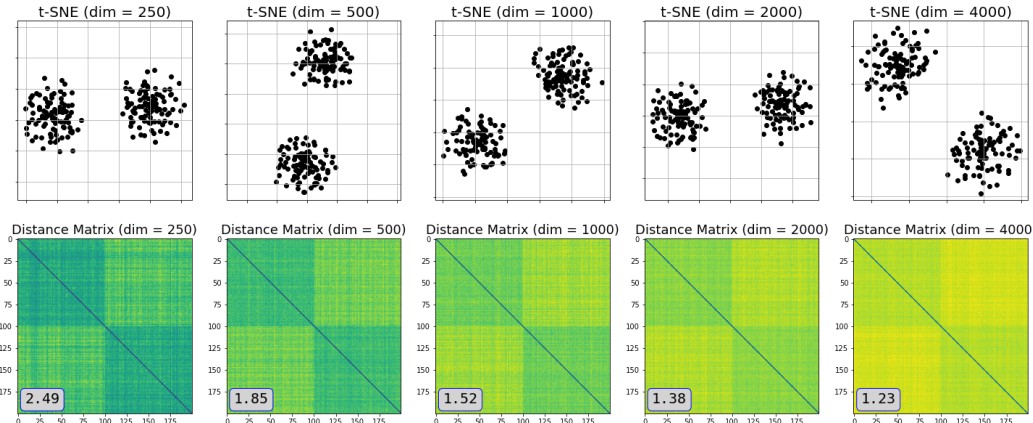

Figure 9: t-SNE's interplay with the concentration of measure phenomenon. The bottom left number in the plots show the aspect ratio (i.e. largest interpoint distance to the smallest interpoint distance).

Figure 10 below shows that the effects of adding a poison point persists even with larger Gaussian clustered datasets.

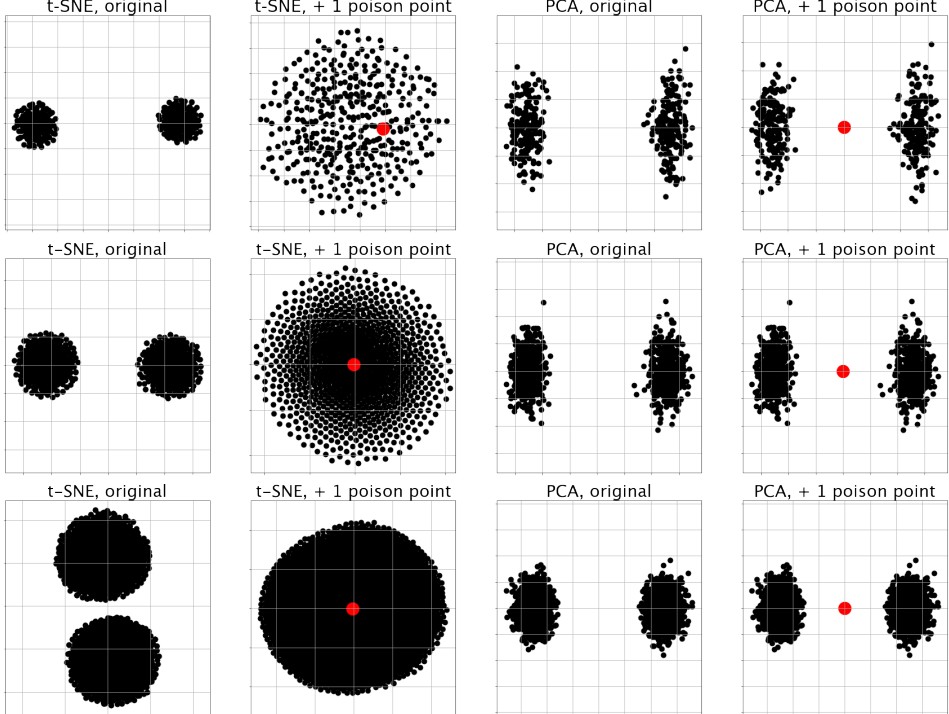

Figure 10: Top row: 400 total points, Middle row: 1000 total points, Bottom Row: 5000 total points

Figure 11 provides an experimental demonstration of Theorem 7.

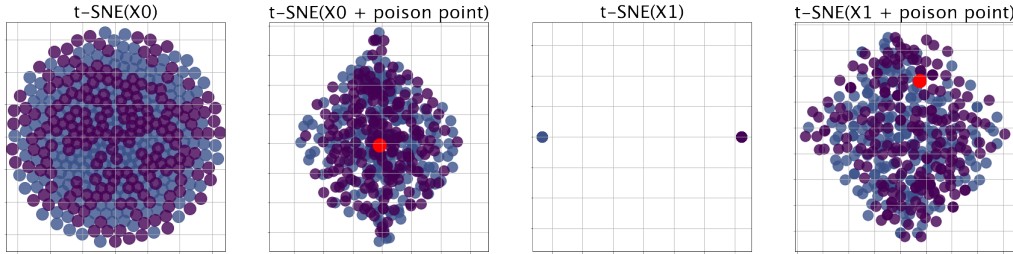

Figure 11: An experimental demonstration of Theorem 7. This shows that $X_0$, the un-clustered input (with silhouette score of 0), has an un-clustered t-SNE output (first panel), while $X_1$, the clustered input (with silhouette score around $0.422$), has a very clustered t-SNE output (third panel). However, with the insertion of the poison point, the t-SNE outputs are similarly un-clustered (second and fourth panels).

## A.3 EMPIRICAL COMPARISON WITH UMAP

It is natural to ask whether related methods like UMAP suffer from the same failure modes that we point out for t-SNE in this work.

In Figure 12, we see how UMAP's output visualization is qualitatively unchanged between the given single cell dataset and its near-simplex impostor. However, unlike t-SNE, it does not display exact additive invariance, as the positioning of clusters changes. In Figure 13, we observe that UMAP, like t-SNE, displays extreme instability near the unit simplex. In Figure 14, we observe that UMAP's response to a single poison point is not as pronounced as that of t-SNE, but the injection of multiple poison points can still change the shape of the clusters significantly.

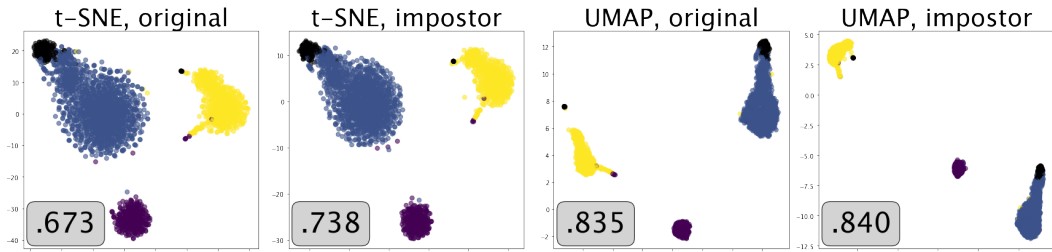

Figure 12: UMAP on a real single-cell dataset versus (near simplex) impostor. The datasets and cluster partitions are those used in Figure 2. Note that the bottom left number in each panel is the silhouette score of the embedding with respect to the clustering.

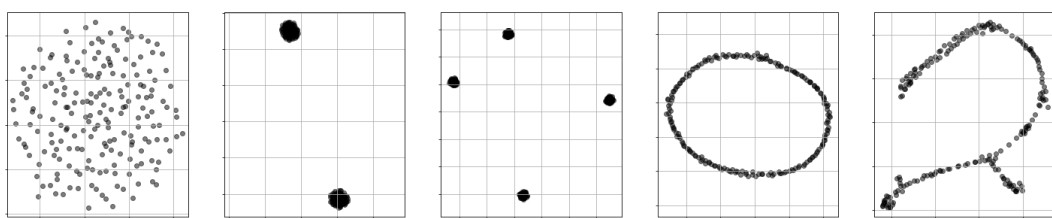

Figure 13: Various different 2D UMAP visualizations produced by small adversarial perturbations of a 200-point unit regular simplex. These are the same datasets visualized in Figure 3.

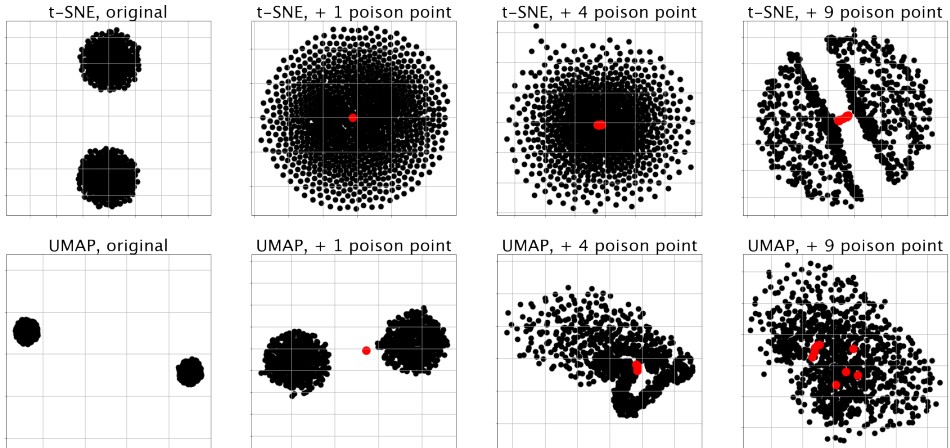

Figure 14: A comparison of t-SNE and UMAP's response to poison points, the original input being 1000 points sampled from a mixture of two Gaussians in 2000 dimensions. Notably, t-SNE, unlike UMAP, seems to have the property that even for very large datasets, a single poison point can destroy the cluster structure. UMAP generally has quite diverse responses to poison points.

## A.4  CALINSKI-HARABASZ INDEX

For an $n$-point dataset $X = \{x_1, \ldots, x_n\} \subset \mathbb{R}^{n-1}$ and a partition of the dataset into non-empty clusters $C_1 \sqcup C_2 \sqcup \cdots \sqcup C_k = [n]$ with $n > k > 1$, the Calinski-Harabasz Index is defined as the ratio of the distance between cluster centers to the internal distance to a cluster's center. Let $E$ be the function from $S \subseteq [n]$ to $\mathbb{R}^{n-1}$ such that:

$$E(S) = \frac{1}{|S|} \sum_{i \in S} x_i.$$

Then the Calinski-Harabasz Index is[6]:

$$\mathrm{CH}(X; C_{m \in [k]}) = \frac{\frac{1}{k-1} \sum_{m \in [k]} |C_m| \cdot \|E(C_m) - E([n])\|^2}{\frac{1}{n-k} \sum_{m \in [k]} \sum_{i \in C_m} \|x_i - E(C_m)\|^2}.$$

It ranges from 0 to $\infty$ with a score of $\infty$ being assigned to perfectly clustered data, 1 to unclustered data and 0 to incorrectly clustered data.

Now we provide an analogue to Theorem 3 with respect to the Calinski-Harabasz Index:

**Theorem 10.** *Fix any $n > k > 1$, and $n$-point dataset $X \subset \mathbb{R}^{n-1}$ with partition $C_1 \sqcup \cdots \sqcup C_k = [n]$ such that $\mathrm{CH}(X; C_{m \in [k]}) > 1$. For all $1 < \epsilon \leq \mathrm{CH}(X; C_{m \in [k]})$, there exists $n$-point dataset $X_\epsilon \subset \mathbb{R}^{n-1}$ such that*

$$\mathrm{CH}(X_\epsilon; C_{m \in [k]}) = \epsilon,$$

*yet, for any $\rho \in [1, n-1]$:*

$$\text{t-SNE}_\rho(X) = \text{t-SNE}_\rho(X_\epsilon).$$

**Corollary 11.** *For all $n \geq 4$ even, and partition $C_1 \sqcup C_2 = [n]$ such that $|C_1| = |C_2| = \frac{n}{2}$. There exist a sequence of $n$-point datasets in $\mathbb{R}^{n-1}$, $\{X_\epsilon\}_{1 < \epsilon \leq \infty}$, with*

$$\mathrm{CH}(X_\epsilon; C_1, C_2) = \epsilon$$

*such that for any $\rho \in [1, n-1]$, $\bigcap_{1 < \epsilon \leq \infty} \text{t-SNE}_\rho(X_\epsilon)$ contains $n$-point dataset $Y \subseteq \mathbb{R}^2$ with*

$$\mathrm{CH}(Y; C_1, C_2) = \infty.$$

---

[6]If the denominator and numerator are 0, then $\mathrm{CH}(X; C_{m \in [k]}) := 1$. If only the denominator is 0, then $\mathrm{CH}(X; C_{m \in [k]}) := \infty$.

**Proof of Theorem 10.** First, let us assume that $\text{CH}(g(C); C_{m \in [k]}) < \infty$. Let $g$ be the function from Corollary 19, and $f(C) = \text{CH}(g(C); C_{m \in [k]})$. Note that $f$ is continuous whenever the denominator of $\text{CH}(\cdot; C_{m \in [k]})$ is non-zero which is always the case for $C \in [0, 1]$. Therefore, the image of $f$ on $[0, 1)$ contains the interval $(f(1), f(0)] = (1, \text{CH}(X; C_{m \in [k]})]$. Thus, for all $\epsilon \in (1, \text{CH}(X; C_{m \in [k]})]$, there exists $C \in [0, 1)$ such that $X_\epsilon = g(C)$ satisfies the hypothesis.

If $\text{CH}(g(C); C_{m \in [k]}) = \infty$, then $f$ is continuous on $(0, 1)$ only. Thus for all $\epsilon \in (1, \text{CH}(X; C_{m \in [k]}))$, there exists $C \in (0, 1)$ such that $X_\epsilon = g(C)$ satisfies the hypothesis and for $\epsilon = \text{CH}(X; C_{m \in [k]}))$, $X_\epsilon = X$ satisfies the hypothesis. □

For proof of Corollary 11 see Appendix A.6.

A.5 DUNN INDEX

For an $n$-point dataset $X = \{x_1, \ldots, x_n\} \subset \mathbb{R}^{n-1}$ and a partition of the dataset into clusters $C_1 \sqcup C_2 \sqcup \cdots \sqcup C_k = [n]$ with $|C_{m \in [k]}| > 1$, the Dunn index measures the ratio between the minimum inter-cluster distance and maximum intra-cluster distance. Specifically, the Dunn index is given by the expression[7]

$$\text{DI}(X; C_{m \in [k]}) = \frac{\min_{m, l \in [k], m \neq l, i \in C_m, j \in C_l} \|x_i - x_j\|}{\max_{m \in [k], i, j \in C_m} \|x_i - x_j\|}.$$

It ranges from 0 to $\infty$ with a score of 0 being assigned to incorrectly clustered data, 1 to unclustered data, and $\infty$ to perfectly clustered data.

Now we restate Theorem 3 with respect to the Dunn Index:

**Theorem 12.** *Fix any $n > k > 1$, and $n$-point dataset $X \subset \mathbb{R}^{n-1}$ with partition $C_1 \sqcup \cdots \sqcup C_k = [n]$ such that $|C_{m \in [k]}| > 1$ and $\text{DI}(X; C_{m \in [k]}) > 1$. For all $1 < \epsilon \leq \text{DI}(X_\epsilon; C_{m \in [k]})$, there exists $n$-point dataset $X_\epsilon \subset \mathbb{R}^{n-1}$ such that*

$$\text{DI}(X_\epsilon; C_{m \in [k]}) = \epsilon,$$

*yet, for any $\rho \in [1, n-1]$:*

$$\text{t-SNE}_\rho(X) = \text{t-SNE}_\rho(X_\epsilon).$$

**Corollary 13.** *For all $n \geq 4$ even, and partition $C_1 \sqcup C_2 = [n]$ such that $|C_1| = |C_2| = \frac{n}{2}$. There exist a sequence of $n$-point datasets in $\mathbb{R}^{n-1}$, $\{X_\epsilon\}_{1 < \epsilon \leq \infty}$, with*

$$\text{DI}(X_\epsilon; C_1, C_2) = \epsilon$$

*such that for any $\rho \in [1, n-1]$, $\bigcap_{1 < \epsilon \leq \infty} \text{t-SNE}_\rho(X_\epsilon)$ contains $n$-point dataset $Y \subseteq \mathbb{R}^2$ with*

$$\text{DI}(Y; C_1, C_2) = \infty.$$

**Proof of Theorem 12.** Let $g$ be the function from Corollary 19, and $f(C) = \text{DI}(g(C); C_{m \in [k]})$. Fix $i, j \in [n]$ such that:

$$\min_{m, l \in [k], m \neq l, i' \in C_m, j' \in C_l} \|x_{i'} - x_{j'}\| = \|x_i - x_j\|,$$

and $t, r \in [n]$ such that:

$$\max_{m \in [k], i', j' \in C_m} \|x_{i'} - x_{j'}\| = \|x_r - x_t\|,$$

Then:

$$f(C) = \frac{\sqrt{(1 - C) \cdot \|x_i - x_j\| + C}}{\sqrt{(1 - C) \cdot \|x_r - x_t\| + C}}.$$

Thus the image of $f$ on $[0, 1)$ is $(f(0), f(1)] = (1, \text{DI}(X; C_{m \in [k]})]$. Therefore, for all $\epsilon \in (1, \text{DI}(X; C_{m \in [k]})]$, there exists $C \in [0, 1)$ such that $X_\epsilon = g(C)$ satisfies the hypothesis. □

For proof of Corollary 13 see Appendix A.6.

---

[7]If the denominator and numerator are 0, then $\text{DI}(X; C_{m \in [k]}) := 1$. If only the denominator is 0, then $\text{DI}(X; C_{m \in [k]}) := \infty$.

A.6 OMITTED PROOFS

The main effort of this section will be to prove Lemma 6, which morally gives us Theorem 3 and Theorem 5. In order to do this, we first introduce a number of technical lemmas which collectively show that t-SNE is invariant under additive and multiplicative scaling of the input.

**Lemma 14.** *Let $H(\cdot)$ denote the entropy function. For any $n > 2$, $X = \{x_1, \ldots, x_n\} \subset \mathbb{R}^{n-1}$ and $\rho \in (1, n-1)$, there is a unique $\sigma_i \geq 0$ that minimizes*

$$\left| H(P_{\cdot|i}(X; \sigma_i)) - \log_2 \rho \right|.$$

*Proof.* This follows easily from the fact that $H(P_{\cdot|i}(X; \sigma))$ is a continuous, strictly increasing function of $\sigma$ (see e.g. Lemma 4.2 of Jeong & Wu (2024)), where $\lim_{\sigma \to \infty} H(P_{\cdot|i}(X; \sigma)) = \log_2(n-1)$ and $H(P_{\cdot|i}(X; 0)) \in [0, \log_2(n-1)]$. $\square$

**Definition 15.** *For any $n \geq 1$, dataset $X = \{x_1, \ldots, x_n\} \subset \mathbb{R}^{n-1}$, and $C \geq 0$, define $X_{+C} = \{x'_1, \ldots, x'_n\} \subset \mathbb{R}^{n-1}$ such that for all $i \neq j$*

$$\|x'_i - x'_j\|^2 = \|x_i - x_j\|^2 + C.$$

**Lemma 16.** *Fix any $n \geq 1$. For all $n$-point datasets $X = \{x_1, \ldots, x_n\} \subset \mathbb{R}^{n-1}$ and $C \geq 0$, there exists $X_{+C} = \{x'_1, \ldots, x'_n\} \subset \mathbb{R}^{n-1}$ such that for all $i \neq j$, $\|x'_i - x'_j\|^2 = \|x_i - x_j\|^2 + C$.*

*Proof.* Let $D$ be the inter-point squared distance matrix of $X$. Thus, the inter-point squared distance matrix of $X_{+C}$ is $D_{+C} = D + C \cdot (11^T - I_n)$. By a famous theorem by Schoenberg (1938), $X_{+C}$ is isometrically embeddable in $\mathbb{R}^{n-1}$ with respect to $\ell_2$ metric if and only if $\forall u \in \mathbb{R}^n$ with $u^T \vec{1} = 0$, $u^T D_{+C} u \leq 0$ holds. Indeed,

$$u^T D_{+C} u = u^T D u + C \cdot (u^T \vec{1})(\vec{1}^T u) - C \cdot u^T u = u^T D u - C \cdot \|u\|^2 \leq 0,$$

where the final inequality uses the fact that $D$ is embeddable. $\square$

**Lemma 17.** *Fix any $n \geq 2$. For all $n$-point datasets $X \subset \mathbb{R}^{n-1}$, $\rho \in [1, n-1]$, and $C \geq 0$:*

$$\text{t-SNE}_\rho(X) = \text{t-SNE}_\rho(X_{+C}).$$

*Proof.* It is sufficient to show that the input affinity matrices for $X$ and $X_{+C}$ are identical. Indeed, for all $i, j \in [n], i \neq j$:

$$P_{i|j}(X) = \frac{\exp\left(-\|x_i - x_j\|_2^2/(2\sigma_i^2)\right)}{\sum_{k \neq i} \exp\left(-\|x_i - x_k\|_2^2/(2\sigma_i^2)\right)}$$

$$= \frac{\exp\left(-(\|x_i - x_j\|_2^2 + C)/(2\sigma_i^2)\right)}{\sum_{k \neq i} \exp\left(-(\|x_i - x_k\|_2^2 + C)/(2\sigma_i^2)\right)} = P_{i|j}(X_{+C}).$$

$\square$

**Lemma 18.** *Fix any $n \geq 2$. For all $n$-point datasets $X \subset \mathbb{R}^{n-1}$, $\rho \in (1, n-1)$, and $C > 0$:*

$$\text{t-SNE}_\rho(X) = \text{t-SNE}_\rho(C \cdot X).$$

*Proof.* First note that for any dataset $X$ and its scaling $C \cdot X$, and all $\sigma_i \geq 0$, we have the following:

$$P_{j|i}(C \cdot X; C \cdot \sigma_i) = \frac{\exp(-C^2 \cdot \|x_i - x_j\|^2/(2C^2 \cdot \sigma_i^2))}{\sum_{k=1, k \neq j}^n \exp(-C^2 \cdot \|x_i - x_k\|^2/(2C^2 \cdot \sigma_i^2))} = P_{j|i}(X; \sigma_i).$$

Let $H(\cdot)$ denote the entropy function. By the above, $H(P_{\cdot|i}(X; \sigma_i)) = H(P_{\cdot|i}(C \cdot X; C \cdot \sigma_i))$. Let $\sigma_i^*$ and correspondingly $\gamma_i^*$ be the (unique, per Lemma 14) neighborhood scalings that satisfy the perplexity condition for $X$ and $C \cdot X$ respectively (see Section 3). Then $\gamma_i^* = C \cdot \sigma_i^*$.

Therefore $P_{\cdot|i}(C \cdot X; \gamma_i^*) = P_{\cdot|i}(C \cdot X; C \cdot \sigma_i^*) = P_{\cdot|i}(X; \sigma_i^*)$, yielding the result. $\square$

The above lemmas give us the following useful corollary that will allow us to prove Lemma 6, Theorem 3, Theorem 10, and Theorem 12.

**Corollary 19.** *Fix any $n \geq 2$, and $X = \{x_1, \ldots, x_n\} \subset \mathbb{R}^{n-1}$. There exists a well-defined, continuous function $g : [0, 1] \to \mathbb{R}^{n \times n-1}$ such that:*

$$C \mapsto ((1 - C) \cdot X)_{+C},$$

*and for all $\rho \in [1, n - 1]$ and $C \in [0, 1) :$*

$$\text{t-SNE}_\rho(X) = \text{t-SNE}_\rho(g(C)).$$

*Proof.* $g$ is well defined by Lemma 16 and WLOG continuous since it continuously transforms the distances in $X$:

$$\forall i, j \in [n], i \neq j, \qquad \|g(C)_i - g(C)_j\| = \sqrt{(1 - C)^2 \cdot \|x_i - x_j\|^2 + C}.$$

Moreover, by Lemmas 17 and 18, for all $C \in [0, 1) :$

$$\text{t-SNE}_\rho(X) = \text{t-SNE}_\rho(g(C)).$$

$\square$

Using the additive and multiplicative invariance of t-SNE, we now prove Lemma 6:

***Proof of Lemma 6.*** Fix any $\epsilon > 0$. It suffices to show that $\text{Im}(\text{t-SNE}_{\rho,n}) \subseteq \text{t-SNE}_\rho(\Delta_\epsilon)$. Fix any $Y \in \text{Im}(\text{t-SNE}_{\rho,n})$, there exists a $n$-point dataset $X = \{x_1, \ldots, x_n\} \subset \mathbb{R}^{n-1}$ such that:

$$Y \in \text{t-SNE}_\rho(X).$$

Using additive and multiplicative invariance, we will manipulate $X$ such that it is in $\Delta_\epsilon$ which by Lemma 17 and Lemma 18 will not change the output. With respect to $X$, define $g : [0, 1] \to \mathbb{R}^{n \times n-1}$ as in Corollary 19. Then for all $C \in [0, 1) :$

$$\text{t-SNE}_\rho(X) = \text{t-SNE}_\rho(g(C)).$$

By the construction of $g$, as $C$ approaches 1, all inter-point distances in $g(C)$ approach 1. Thus there must exist some $C \in [0, 1)$ such that $g(C) \in \Delta_\epsilon$ which completes the proof. $\square$

Using the above lemmas, Theorem 3, Corollary 4, and Theorem 5 are proven.

**Theorem 3.** *Fix any $n > k > 1$, and $n$-point dataset $X \subset \mathbb{R}^{n-1}$ with partition $C_1 \sqcup \cdots \sqcup C_k = [n]$ such that $|C_{m \in [k]}| > 1$ and $\bar{\mathcal{S}}(X; C_{m \in [k]})$ is well defined. For all $0 < \epsilon \leq 1$, there exists $n$-point dataset $X_\epsilon \subset \mathbb{R}^{n-1}$ such that*

$$\bar{\mathcal{S}}(X_\epsilon; C_{m \in [k]}) = \epsilon \cdot \bar{\mathcal{S}}(X; C_{m \in [k]}),$$

*yet, for any $\rho \in [1, n - 1]$,*

$$\text{t-SNE}_\rho(X) = \text{t-SNE}_\rho(X_\epsilon).$$

***Proof of Theorem 3.*** Let $g$ be the function from Corollary 19, and $f(C) = \bar{\mathcal{S}}(g(C); C_{m \in [k]})$. Note that $f$ is continuous for $C \in [0, 1]$ since $g$ is continuous, and $\bar{\mathcal{S}}(\,\cdot\,; C_{m \in [k]})$ is continuous whenever for all $i \in [n], a(i), b(i) \neq 0$ which follows from $\bar{\mathcal{S}}(X; C_{m \in [k]})$ being well-defined and the definition of $g$. Therefore, the image of $f$ on $[0, 1)$ contains the interval $(f(1), f(0)] = (0, \bar{\mathcal{S}}(X; C_{m \in [k]})]$ (or if $\bar{\mathcal{S}}(X; C_{m \in [k]}) \leq 0, [\bar{\mathcal{S}}(X; C_{m \in [k]}), 0))$. Thus, for all $\epsilon \in (0, 1]$, there exists $C \in [0, 1)$ such that $X_\epsilon = g(C)$ satisfies the hypothesis. $\square$

Now we can prove Corollary 4, Corollary 11, and Corollary 13 simultaneously:

***Proof of Corollaries 4, 11, and 13.*** The proof proceeds by showing a dataset and its output who have an average silhouette score of 1, Calinski-Harabasz index of $\infty$, and Dunn index of $\infty$, and then applies Theorem 3, Theorem 10, and Theorem 12 respectively. WLOG fix partition $C_1 \sqcup C_2 = [n]$ with $C_1 = [1, n/2]$ and $C_2 = [n/2 + 1, n]$. Consider the $n$-point dataset, $X = \{x_1, \ldots, x_n\} \subset \mathbb{R}^{n-1}$, such that for all $i \in C_1$, $x_i = \vec{0}$, and for all $i \in C_2$, $x_i = \vec{e_1}$.

Routine calculations show that the conditional input affinities are:

$$
P_{i|j} = \begin{cases} \frac{1}{\frac{n}{2} - 1 + \frac{n}{2} \exp\left(-\frac{1}{2\sigma_j^2}\right)} & i \in C(j), i \neq j \\[4mm] \frac{\exp\left(-\frac{1}{2\sigma_j^2}\right)}{\frac{n}{2} - 1 + \frac{n}{2} \exp\left(-\frac{1}{2\sigma_j^2}\right)} & i \notin C(j) \\[4mm] 0 & i = j. \end{cases}
$$

By symmetry, $\sigma_j = \sigma_i$ for all $i, j \in [n]$. Hence, let $\sigma$ be the neighborhood size for all $j \in [n]$ which is non-zero and well defined for $\rho \in [1, n-1]$. Thus the symmetrized input affinities are:

$$
P_{ij} = \begin{cases} \frac{1}{\frac{n^2}{2} - n + \frac{n^2}{2} \exp\left(-\frac{1}{2\sigma^2}\right)} & i \in C(j), i \neq j \\[4mm] \frac{\exp\left(-\frac{1}{2\sigma^2}\right)}{\frac{n^2}{2} - n + \frac{n^2}{2} \exp\left(-\frac{1}{2\sigma^2}\right)} & i \in C_1, j \in C_2 \\[4mm] 0 & i = j. \end{cases}
$$

Any set $Y = \{y_1, \ldots, y_n\} \subset \mathbb{R}$ is a global minimizer if $P_{ij} = Q_{ij}$ for all $i, j \in [n]$. In this case, this is achieved if $y_{i \in C_1} = 0$ and $y_{i \in C_2} = \sqrt{\exp\left(\frac{1}{2\sigma^2}\right) - 1}$. Furthermore, since $Y$ can be isometrically embedded in $\mathbb{R}^d$ for all $d \geq 1$, this result holds for t-SNE embeddings of all dimensions.

To finish the proof note that for all $i \in [n]$ $a(i) = 0$ when defined with respect to $Y$ and partition $C_1 \sqcup C_2$. $\qquad\square$

**Theorem 5.** *Fix any $n \geq 2$ and $\rho \in [1, n-1]$. For all $\epsilon > 0$ and all $Y, Y' \in \mathsf{Im}(\text{t-SNE}_{\rho,n})$, there exists $n$-point datasets $X = \{x_1, \ldots, x_n\}$ and $X' = \{x'_1, \ldots, x'_n\} \subset \mathbb{R}^{n-1}$ such that $\forall i \neq j$*

$$
1 - \epsilon \leq \frac{\|x_i - x_j\|^2}{\|x'_i - x'_j\|^2} \leq 1 + \epsilon,
$$

*yet $Y \in \text{t-SNE}_\rho(X)$ and $Y' \in \text{t-SNE}_\rho(X')$.*

***Proof of Theorem 5.*** The proof is immediate by application of Lemma 6. $\qquad\square$

**Theorem 7.** *Take $n \geq 4$ even. There exists two $n$-point datasets such that:*

- *$X_0 \subset \mathbb{R}^{n-1}$ has silhouette score $0$ with respect to all partitions, and*

- *$X_1 \subset \mathbb{R}^{n-1}$ has silhouette score $\frac{\sqrt{3}-1}{\sqrt{3}} \geq 0.42$ with respect to some partition of points,*

*yet there exists $x_0 \in \mathbb{R}^{n-1}$ such that for $\rho = 1$, $\text{t-SNE}_\rho(X_0 \cup x_0) = \text{t-SNE}_\rho(X_1 \cup x_0)$.*

***Proof of Theorem 7.*** Let $J_n := \vec{1}_n \vec{1}_n^T$. To show the existence of $X_0$ and $X_1$, it suffices to write down their squared Euclidean interpoint distance matrices. Denote these $D^2(X_0)$ and $D^2(X_1)$ respectively. Define
$$
D^2(X_0) := J_n - I_n.
$$
This is a squared Euclidean distance matrix because it can be realized by a regular simplex (all points are equidistant). Now define

$$
D^2(X_1) := \begin{bmatrix} J_{n/2} - I_{n/2} & (1+r)J_{n/2} \\ (1+r)J_{n/2} & J_{n/2} - I_{n/2} \end{bmatrix}.
$$

This is also a squared Euclidean distance matrix because one can start with an embedding in $\mathbb{R}$ where $x_1 = \cdots = x_{n/2} = 0$ and $x_{n/2+1} = \cdots = x_n = \sqrt{r}$, which has squared Euclidean distance matrix profile of

$$\tilde{D}^2 = \begin{bmatrix} \vec{0}_{n/2}\vec{0}_{n/2}^T & rJ_{n/2} \\ rJ_{n/2} & \vec{0}_{n/2}\vec{0}_{n/2}^T \end{bmatrix},$$

and then, recalling that the set of squared Euclidean distance matrices form a cone, one can argue that $D^2(X_1) = \tilde{D}^2 + D^2(X_0)$ is also squared Euclidean.

For any partition of $[n]$, it is clear that the silhouette score of $X_0$ with respect to that partition is zero since all points are equidistant. As for $X_1$, take $r = 2$ and observe that for $C_1 = \{1, \ldots, n/2\}$ and $C_2 = \{n/2 + 1, \ldots, n\}$, we have $\bar{S}(X_1; C_1, C_2) = \frac{\sqrt{3}-1}{\sqrt{3}} \approx 0.422$.

Let $X_0$ and $X_1$ be Euclidean vector realizations of $D^2(X_0)$ and $D^2(X_1)$ each centered at the origin. Let $x_0$ be the origin, and hence the mean of both datasets. Note that $x_0$ will be equidistant to all the points in $X_0$, and likewise with $X_1$.

To compute the distance of the mean $x_0$ to other points, we make use of the fact that for all $i \in [n]$

$$\|x_0 - x_i\|^2 = \frac{1}{n} \sum_{j=1}^n \|x_0 - x_j\|^2 = \frac{1}{2} \cdot \frac{1}{n^2} \sum_{j=1}^n \sum_{k=1}^n \|x_k - x_j\|^2.$$

The resulting Euclidean distance matrices are as follows (where the distances corresponding to the appended point are written in the last row and column).

$$D^2(X_0 \cup x_0) = \begin{bmatrix} J_n - I_n & (\frac{1}{2} - \frac{1}{2n})\vec{1}_n \\ (\frac{1}{2} - \frac{1}{2n})\vec{1}_n^T & 0 \end{bmatrix},$$

$$D^2(X_1 \cup x_0) = \begin{bmatrix} J_{n/2} - I_{n/2} & (1+r)J_{n/2} & (\frac{1}{2} + \frac{r}{4} - \frac{1}{2n})\vec{1}_{n/2} \\ (1+r)J_{n/2} & J_{n/2} - I_{n/2} & (\frac{1}{2} + \frac{r}{4} - \frac{1}{2n})\vec{1}_{n/2} \\ (\frac{1}{2} + \frac{r}{4} - \frac{1}{2n})\vec{1}_{n/2}^T & (\frac{1}{2} + \frac{r}{4} - \frac{1}{2n})\vec{1}_{n/2}^T & 0 \end{bmatrix}.$$

The key observation is that in both cases (recall the choice of $r = 2$), for all $i, j \in [n]$, $\|x_i - x_0\|^2 < \|x_i - x_j\|^2$. By Lemma 20, this implies that for $\rho = 1$, $P_{i|j}(X_0) = P_{i|j}(X_1) = 0$ for all $i, j \in [n]$. In turn, $P_{ij}(X_0) = P_{ij}(X_1) = 0$. Meanwhile $P_{i|0}(X_0) = P_{i|0}(X_1) = 1/n$ (since $x_0$ is equidistant from all other points) and $P_{0|i}(X_0) = P_{0|i}(X_0) = 1$ for all $i \in [n]$ (since $x_0$ is every point's nearest neighbor). Thus both cases yield the same $P$ matrix:

$$P(X_0 \cup x_0) = P(X_1 \cup x_0) = \frac{1}{2(n+1)} \begin{bmatrix} \vec{0}_n\vec{0}_n^T & (1 + \frac{1}{n})\vec{1}_n \\ (1 + \frac{1}{n})\vec{1}_n^T & 0 \end{bmatrix}.$$

The equivalence of their loss landscapes follows immediately. $\qquad \square$

**Lemma 20.** *For $\rho = 1$ and any $X = \{x_1, \ldots, x_n\} \subset \mathbb{R}^D$,*

$$P_{i|j}(X; \sigma_j^*) = \frac{1}{|N(j)|} \cdot \boldsymbol{I}[i \in N(j)] \quad where \quad N(j) := \underset{k \in [n] \setminus \{j\}}{\arg \min} \|x_k - x_j\|$$

*Proof.* Note that $\rho = 1$ corresponds to $H(P_{\cdot|j}) = 0$, so we optimize the neighborhood size $\sigma_j$ to minimize the entropy of the conditional affinities. This is done by taking $\sigma_j = 0$. Recall

$$P_{i|j}(X; 0) = \lim_{\sigma_j \to 0} \frac{\exp(-\|x_i - x_j\|^2/2\sigma_j^2)}{\sum_{k=1, k \neq j}^n \exp(-\|x_k - x_j\|^2/2\sigma_j^2)}$$

In the limit, only those $x_k$ which are closest to $x_j$ will dominate, i.e. $x_k$ where $k \in N(j)$. $\qquad \square$

A.7 IMPOSTOR DATASET CONSTRUCTION

The construction of an impostor dataset based on an input dataset is done as follows.

---

**Algorithm 1** Impostor Dataset Creation

---

**Require:** Dataset $X = \{x_1, \ldots, x_n\}$ with at least two distinct points, and tolerance $\epsilon > 0$
1: Construct squared interpoint distance matrix of $X$, denote it by $D$
2: Form $D' \leftarrow \frac{\epsilon}{\max_{i,j} D_{ij}} \cdot D + (11^\top - I_n)$
3: Run classical multidimensional scaling on $D'$ to obtain its Euclidean embedding

$$X_\epsilon = \{x'_1, \ldots, x'_n\} \subset \mathbb{R}^{n-1}.$$

4: **return** $X_\epsilon$

---

# B APPENDIX: MISREPRESENTATION OF OUTLIERS

## B.1 ADDITIONAL EXPERIMENTS

We provide a comparison of t-SNE and PCA on the BBC news dataset. For ease of presentation, we take a three-cluster, $(n = 1204)$-size subset (business, sports, tech) and we analyze what happens under injection of 120 poison points versus 120 far outliers.

In both Figure 6 and Figure 15, poison points are picked as follows: first we run a $k$-means algorithm on the original dataset; then, for each poison point, we pick one of the these means and 10 random points of the dataset, and we average these two quantities (the idea is to connect the points in a way that contradicts the ground-truth three-clustering). We found $k = 2$ worked well. We pick outlier points as normal vectors centered at the mean of the dataset with variance 32 (the diameter of the original dataset is roughly 1.5).

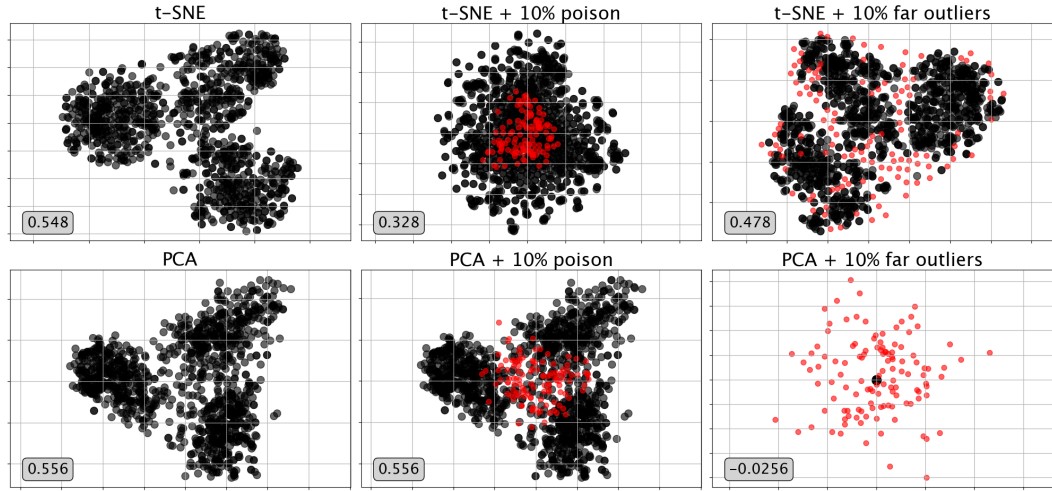

BBC News (transformer word embeddings)

Figure 15: t-SNE vs. PCA on poison points versus outlier points on a three-cluster subset of the BBC news dataset. The label on the bottom left is silhouette score of the plot (sans injected points) with respect to their ground-truth labels.

Below are additional experiments demonstrating t-SNE and PCA visualizations have divergent responses for single poison point injection even as the size of the dataset increases.

### B.2 EMPIRICAL COMPARISON WITH UMAP

UMAP exhibits similar behavior to t-SNE when it comes to depicting far-away outliers. The main difference seems to be that UMAP tends to incorporate outliers deep into clusters, while t-SNE keeps them on the edge of clusters. As seen in Figure 16 (right), this behavior can compromise the quality of clustering.

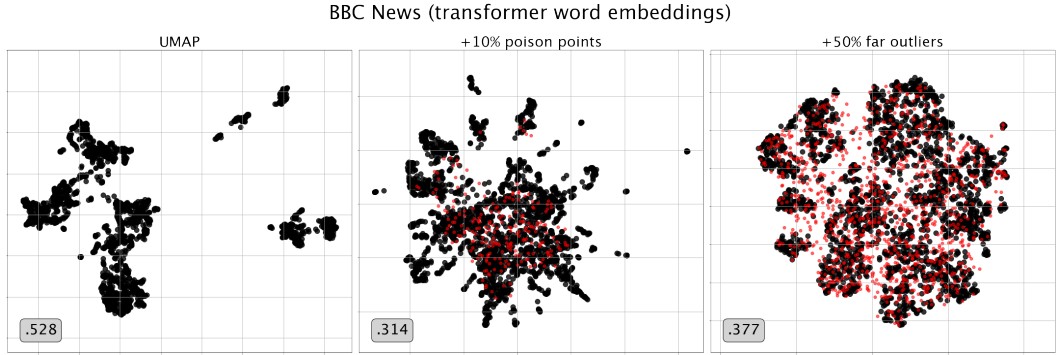

Figure 16: UMAP in response to poison points versus $\alpha$-outliers on the BBC News dataset. The number in the bottom left of each panel is silhouette score with respect to the ground-truth. Compare with Figure 6.

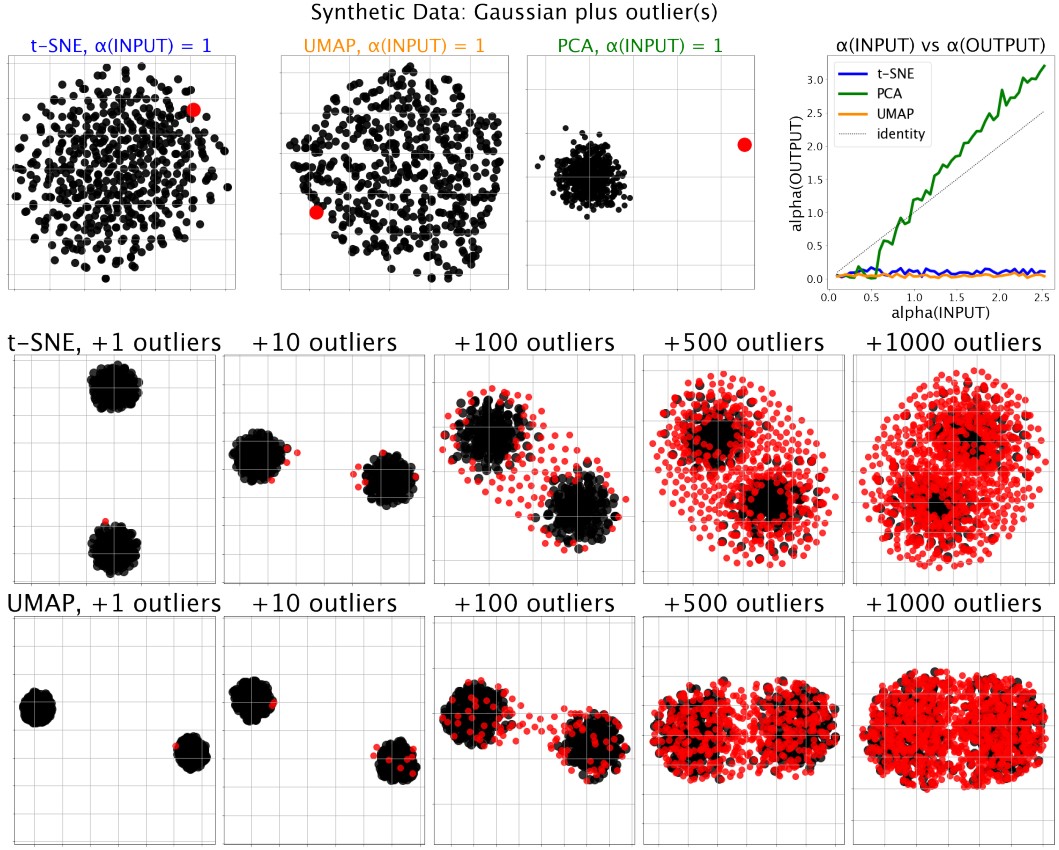

Figure 17: UMAP, like t-SNE, consistently suppresses faraway outliers. One salient difference between t-SNE and UMAP is that UMAP is more inclined to place outliers within clusters, whereas t-SNE often places them at the edge. Compare with Figure 5.

### B.3 OMITTED PROOFS

**Lemma 21.** *Fix* $n \geq 2$ *and* $Y = \{y_0, \ldots, y_{n-1}\} \subset \mathbb{R}^d$. *Let* $\beta := \operatorname{diam}(Y \setminus \{y_0\})$ *and* $\gamma := \min_{j \in [n]} \|y_0 - y_j\|$. *Then*

$$\sum_{i=1}^{n} Q_{0i} \leq \frac{1}{2 + (n-2) \cdot \frac{1+\gamma^2}{1+\beta^2}}.$$

*Proof.* Let $Z_0 = \sum_{i=1}^{n-1} \frac{1}{1+\|y_i-y_0\|^2}$ and $Z_{1:n-1} = \sum_{i,j \in [n-1]: i \neq j} \frac{1}{1+\|y_i-y_j\|^2}$. Then

$$\sum_{i=1}^{n-1} Q_{0i} = \frac{Z_0}{2Z_0 + Z_{1:n-1}} = \frac{1}{2 + Z_{1:n-1}/Z_0}.$$

Now observe that

$$
\begin{aligned}
\frac{Z_{1:n-1}}{Z_0} &= \frac{\sum_{i,j \in [n-1]: i \neq j}(1 + \|y_i - y_j\|^2)^{-1}}{\sum_{j=1}^{n-1}(1 + \|y_0 - y_j\|^2)^{-1}} \\
&\geq \frac{(n-1)(n-2)(1 + \max_{i,j \in [n]} \|y_i - y_j\|^2)^{-1}}{(n-1)(1 + \min_{j \in [n]} \|y_0 - y_j\|^2)^{-1}} \\
&= \frac{(n-2)(1 + \gamma^2)}{1 + \beta^2}.
\end{aligned}
$$

Plugging this back into the previous equation gives the statement. $\qquad \square$

**Lemma 22.** *Fix* $n \geq 2$ *and* $Y = \{y_0, y_1, \ldots, y_{n-1}\} \subset \mathbb{R}^d$. *If* $Y$ *is a* $(\alpha, y_0)$-*outlier configuration such that* $\alpha = \alpha(Y)$, *then there exists* $v \in \mathbb{R}^d$ *such that for all* $i \in [n]$:

$$\|y_i - y_0\| \cdot \frac{\alpha}{1+\alpha} \leq (y_i - y_0) \cdot v \leq \|y_i - y_0\|.$$

*Proof.* Fix $i \in [n]$, let $\beta := \operatorname{diam}(Y \setminus \{y_0\})$, and WLOG let $y_0 = 0$. Let $v$ be the unit vector defining the hyperplane as in Definition 8. Then by Cauchy-Schwarz, $(y_i - y_0) \cdot v \leq \|y_i - y_0\|$. To prove the other side of the inequality, we only need to lower bound the cosine of the angle between $y_i - y_0$ and $v$:

$$(y_i - y_0) \cdot v = \|y_i - y_0\| \cdot \cos(\angle(v, y_i)).$$

Since $v$ is the maximum-margin hyperplane between $y_0 = 0$ and $Y \setminus \{y_0\}$, it holds that $u = v \cdot (\alpha \max\{1, \beta\})$ is in the convex hull of $Y \setminus \{y_0\}$. Indeed, $\|u\| = \inf_{y \in \operatorname{conv}(Y \setminus \{y_0\})} \|y\|$. Thus, we know that the closed ball $\overline{B_\beta(u)}$ contains $\operatorname{conv}(Y \setminus \{y_0\})$. Therefore, there exists $t \in \mathbb{R}^d$ such that $\|t\| \leq \beta, u + t = y_i$, and $u \cdot t \geq 0$. Hence

$$\cos(\angle(v, y_i)) = \frac{v \cdot y_i}{\|y_i\|} = \frac{v \cdot (u + t)}{\|u + t\|} \geq \frac{v \cdot u}{\|u\| + \|t\|} \geq \frac{\alpha \max\{1, \beta\}}{\alpha \max\{1, \beta\} + \max\{1, \beta\}} \geq \frac{\alpha}{1+\alpha},$$

completing the proof. $\qquad \square$

**Theorem 9.** *Fix* $n > 2$ *and* $\rho \in [1, n-1]$. *Let* $Y \in \operatorname{Im}(\text{t-SNE}_{\rho,n})$ *be any* *stationary t-SNE embedding. Then we have:*

$$\alpha(Y) \leq 3.266 + o_n(1).$$

*Proof.* Fix $Y = \{y_0, y_1, \ldots, y_{n-1}\} \in \operatorname{Im}(\text{t-SNE}_{\rho,n})$ and define $\gamma = \min_i \|y_i - y_0\|$. WLOG, let $y_0$ be the outlier point and assume $\gamma > 0$, otherwise the hypothesis goes through trivially. Since $Y$

is stationary, $\frac{\partial \mathcal{L}}{\partial y_0} = 0$. Pick $v$ as in Lemma 22 and observe:

$$
\begin{aligned}
0 = \frac{\partial \mathcal{L}}{\partial y_0} \cdot v &= \sum_{i=1}^{n-1} \frac{(P_{i0} - Q_{i0})(y_0 - y_i) \cdot v}{1 + \|y_0 - y_i\|^2} \\
&\leq -\frac{\alpha}{1+\alpha} \sum_{i=1}^{n-1} P_{i0} \frac{\|y_0 - y_i\|}{1 + \|y_0 - y_i\|^2} + \sum_{i=1}^{n-1} Q_{i0} \frac{\|y_0 - y_i\|}{1 + \|y_0 - y_i\|^2} \\
&\leq -\frac{\alpha}{1+\alpha} \frac{\gamma}{1 + (\gamma + \beta)^2} \sum_{i=1}^{n-1} P_{i0} + \frac{\gamma + \beta}{1 + \gamma^2} \sum_{i=1}^{n-1} Q_{i0} \\
&\leq -\frac{\alpha}{1+\alpha} \frac{\gamma}{1 + (\gamma + \beta)^2} \frac{1 + \sum_{i=1}^{n-1} P_{0|i}}{2n} + \frac{\gamma + \beta}{1 + \gamma^2} \frac{1}{2 + (n-2)\frac{1+\gamma^2}{1+\beta^2}}
\end{aligned}
$$

where, in the fourth line, we use Lemma 21 and the fact that $\sum_{i=1}^{n} P_{i|0} = 1$. Multiplying by $\frac{1+\gamma^2}{\gamma+\beta} \cdot \frac{2n}{1 + \sum_{i=1}^{n-1} P_{0|i}} > 0$ and rearranging, we get that:

$$
\begin{aligned}
\frac{\alpha}{1+\alpha} \cdot \frac{1+\gamma^2}{\gamma+\beta} \cdot \frac{\gamma}{1+(\gamma+\beta)^2} &\leq \frac{1}{2 + (n-2) \cdot \frac{1+\gamma^2}{1+\beta^2}} \cdot \frac{2n}{1 + \sum_{i=1}^{n-1} P_{0|i}} \\
&\leq \frac{1+\beta^2}{(n-2)(1+\gamma^2)} \cdot \frac{2n}{1 + \sum_{i=1}^{n-1} P_{0|i}} \\
&= \frac{1+\beta^2}{1+\gamma^2} \cdot \left(1 + \frac{2}{n-2}\right) \cdot \frac{2}{1 + \sum_{i=1}^{n-1} P_{0|i}}.
\end{aligned}
$$

Thus, by definition of $\alpha$-outlier configuration, $\gamma \geq \alpha \cdot \max\{\beta, 1\}$:

$$
\begin{aligned}
\left(1 + \frac{2}{n-2}\right) \cdot \frac{2}{1 + \sum_{i=1}^{n-1} P_{0|i}} &\geq \frac{\alpha}{1+\alpha} \cdot \frac{\gamma}{\gamma+\beta} \cdot \frac{1+\gamma^2}{1+(\gamma+\beta)^2} \cdot \frac{1+\gamma^2}{1+\beta^2} \\
&\geq \frac{\alpha}{1+\alpha} \frac{\gamma^3}{(\gamma+\beta)^3} \frac{1+\gamma^2}{1+\beta^2} \\
&\geq \frac{\alpha}{1+\alpha} \frac{\alpha^3 \max\{\beta,1\}^3}{(\alpha \max\{\beta,1\} + \beta)^3} \frac{1 + \alpha^2 \max\{\beta,1\}^2}{1+\beta^2} \\
&\geq \frac{\alpha}{1+\alpha} \frac{\alpha^3}{(\alpha + \frac{\beta}{\max\{\beta,1\}})^3} \frac{1+\alpha^2}{2} \\
&\geq \frac{\alpha}{1+\alpha} \frac{\alpha^3}{(1+\alpha)^3} \frac{1+\alpha^2}{2} \\
&= \frac{\alpha^4 (1+\alpha^2)}{2(1+\alpha)^4}.
\end{aligned}
$$

Assume $\alpha \geq 3.26$ (or else the hypothesis holds trivially), then the above is lower-bounded by $0.1876\alpha^2$. Solving for $\alpha$ and observing that each $P_{0|i}$ is non-negative yields that:

$$
\alpha \leq \sqrt{\frac{1}{0.1876} \cdot \left(1 + \frac{2}{n-2}\right) \cdot \left(\frac{2}{1 + \sum_{i=1}^{n-1} P_{0|i}}\right)} \leq \sqrt{\frac{2}{0.1876} \cdot \left(1 + \frac{2}{n-2}\right)} = 3.266 + o_n(1).
$$

This concludes the proof. $\qquad\square$

