# OpenReview forum: "t-SNE Exaggerates Clusters, Provably"
_ICLR.cc/2026/Conference — ICLR 2026 Poster_

### Official Review · Reviewer_cJrt · 2025-10-22

**Soundness:** 3
**Presentation:** 2
**Contribution:** 2
**Rating:** 4
**Confidence:** 3

**Summary:**

This paper theoretically demonstrates that the low-dimensional visualizations produced by t-SNE may not faithfully reflect the underlying high-dimensional structure. Bu using quantitative measures such as the Silhouette score, the authors present "existential" cases that contradict common intuitions about cluster representation. The work raises the need for further discussion on the applicability and reliability of t-SNE in practical data analysis.

**Strengths:**

The paper provides a rigorous theoretical analysis of the stationary points of the t-SNE algorithm, which is widely used for visualizing global data structures.
It demonstrates that counter-intuitive mappings can arise during dimensionality reduction. The arguments are logically consistent and mathematically precise. In particular, the paper gives a clear theoretical foundation for phenomena that had previously been observed only empirically—namely, that t-SNE may exaggerate apparent cluster separations.
This theoretical grounding represents a significant contribution to the understanding of t-SNE’s limitations.

**Weaknesses:**

* The paper presents a rigorous theoretical analysis of t-SNE’s limitations but provides limited practical guidance for practitioners. It focuses on stationary embeddings $t\text{-SNE}_{\rho}(X)$, while in practice, t-SNE is often terminated before full convergence. The paper does not examine whether the reported phenomena (e.g., exaggeration or misrepresentation) persist under early-stopping conditions. Including numerical experiments with partially converged optimization would strengthen its practical relevance.
* Although the authors combine theoretical reasoning with synthetic and some real-world datasets, the paper lacks concrete guidance on how users should interpret or adjust their use of t-SNE in light of these findings.
     Readers may find the results intellectually compelling but unclear on how to apply them safely in practice.
* In Section 4 (“Misrepresentation of Cluster Structure”), the paper shows that cluster separability may not be preserved between high- and low-dimensional spaces, yet it does not discuss how distance metrics or preprocessing might affect this issue in exploratory data analysis.
* In Section 5 (“Misrepresentation of Outliers”), the paper explains how additive invariance leads to distorted proximity perception when outliers are present, but it offers no suggestions on how users might mitigate or account for such effects in practice. Moreover, it would be valuable to discuss how such distortions could affect subsequent analytical tasks—for example, whether applying t-SNE before an outlier detection step could degrade detection accuracy.
     If such degradation occurs, should practitioners avoid using t-SNE for preprocessing in anomaly detection pipelines, or are there possible adjustments or alternative approaches that could mitigate this effect?
     Clarifying these implications would enhance the paper’s practical significance and provide more concrete guidance for real-world use cases.
* Overall, while the theoretical results are clear and compelling, the paper leaves a gap between formal analysis and practical application.

**Questions:**

- It would significantly enhance the value of the paper if the authors could revisit each of the points discussed in the "Weakness" section from the perspective of how the theoretical findings could inform or be applied during practical exploratory data analysis.
 In particular, clarifying how practitioners might leverage these results when interpreting or performing t-SNE analyses would make the paper's contribution more actionable and impactful.

<to be corrected>
- P8, l418: In Theorem 8, the expression $3+o(1)$ is presented without specifying the variable with respect to which the asymptotic term is defined. While it is likely meant as $n\to \infty$, an explicit statement would improve clarity, as multiple asymptotic regimes appear in the paper.  Clarifying this would strengthen the mathematical rigor of the main result.

- P15, l763: \max_{i\neq j} → \max{i,j\in[n], i\neq j}

- If you put figures to the paper, they should be titled and captioned.

---

> ### Author Response · Authors · 2025-11-18
>
> **W1 (local optima)**: The data visualization produced by t-SNE at a local optimum represents the best we can hope for from a gradient-based optimization; this is why our theoretical statements focus on local optima. As for empirics, the numerical experiments depicting t-SNE’s failure modes (Figures 1 through 5; 6 and 7 in the Appendix) use standard early stopping, i.e. not at local convergence, which the reviewer is referring to.
>
> **W2-5 and Q1 (practical guidance)**: We thank the reviewer for bringing up the point about the practical guidance one can draw from our results. Frankly, a comprehensive answer to this is still an open line of research. The focus of our work is to rigorously understand t-SNE’s limitations. Our work establishes the following practical guidance: it is impossible to infer (i) the strength of clustering and (ii) severity of outliers from a t-SNE plot alone. Therefore, on these fronts, t-SNE plots must be suitably cross-checked. We will make sure to make this explicit in the discussion.

---

> > ### Comment · Reviewer_cJrt · 2025-11-26
> >
> > I appreciate the theoretical contributions of the paper. However, from a practical standpoint, that is, when actually running the t-SNE algorithm, it would have been helpful to include a discussion of which indicators should be monitored and how many optimization steps are typically required. Such an analysis would have strengthened the practical value of the work.
> >
> > The authors’ response partially addresses my concerns; however, my overall assessment of the paper remains unchanged.

---

### Official Review · Reviewer_WD8K · 2025-10-29

**Soundness:** 3
**Presentation:** 2
**Contribution:** 3
**Rating:** 4
**Confidence:** 5

**Summary:**

This paper is about the faithfulness of t-SNE visualizations to high-dimensional data. In the first part, the authors illustrate counter-intuitive phenomena of high-dimensional data and discusses their effects on t-SNE visualizations. They also present the novel notion of poison points that drastically alter the t-SNE plot. In the second part, they discuss how t-SNE is unable to place outlier points far away in its visualization. Several of the statements are backed by formal proofs and the findings are qualitatively illustrated on both synthetic and real data.

**Strengths:**

- S1 The paper is well-written and for the most part easy to follow. Formal statements are well explained in natural language. Most examples are well-chosen to illustrate the findings.
- S2: The fact that a high-dimensional dataset can be arbitrarily close to a simplex, while still encoding any structure is interesting and deserves broader dissemination.
- S3: The striking effect of poison points (Fig 3) is also very interesting and novel.
- S4: It is impressive that the authors manage to formally back many of their statements, given that t-SNE and related methods are not easy to analyze theoretically.
- S5: The authors provide code, facilitating reproducibility.

**Weaknesses:**

**Major**
*W1 Presentation:* While the paper is accessible, I think the interpretation of high-dimensional data structure needs to be more nuanced. In Table 1 and related figures as well as Fig 1,2 the authors convey that t-SNE can produce structured visualizations (e.g. clusters, or astronaut shapes) of data that does not actually carry this structure (because it has low clustering scores, has nearly uniform distances, and has similar pairwise distances as datasets whose t-SNE embeddings are very different). I think it is critical to discuss in what way this high-dimensional data is structured in more detail.

Currently, the discussion focuses exclusively on the value of the pairwise distances, which are all nearly the same, suggesting a lack of structure. However, other perspectives, e.g., the kNN graph of the data (used by many visualization tools also including PHATE, clustering methods like modularity clustering, and much general manifold learning) would tell precisely the opposite story: The kNN graph would likely encode structure very similar to what t-SNE (which performs a layout of this graph) shows.

The kNN graph is multiplicatively and additively invariant. So the kNN graphs of the 2D embeddings in Fig 2 and of $n-1$-dimensional configurations obtained by Alg 1 from these 2D embeddings are exactly the same (despite these distances satisfying Thm 5 at arbitrarily small $\varepsilon$). So from this point of view, the high-dimensional point clouds of Fig 2 are neither similar nor unstructured, but (a) carry distinct kNN graph structure that is (b) very different from each other.

This perspective could also be made quantitative by popular clustering metrics, like the modularity of the kNN graph, its conductance, or its spectral gap, or by e.g. the performance of a kNN classifier on the high-dimensional data. For instance, on the data with two clusters for Table 1, the modularity of the kNN graph is likely very close to its maximal value of 1. From this perspective, t-SNE does *not* exaggerate clusters. Given the different behavior of distances in high-dimensional spaces, which the authors mention in several places, I wonder if metrics like Silhouette score are even the appropriate choices.

This tension between different interpretations of high-dimensional structure is interesting (as is the authors' proof that one can have essentially *any* kNN-graph structure in an $\varepsilon$-ball around the regular simplex). But currently, the discussion in the paper is very one-sided, only focusing on the distance perspective.

In addition, the above is more about the interpretation of high-dimensional data in general and less about a specific issue with t-SNE. The paper's storyline (including the title) should reflect this. As mentioned above, any method relying on the kNN graph has additive and multiplicative invariance, e.g., Leiden clustering. But would the authors similarly claim that Leiden clustering "exaggerates clusters" by finding the partition that realizes the kNN graph's high modularity?

There is some discussion in this direction starting in line 478, but this is insufficient. The framing is my main issue. I am happy to increase my score if it gets suitably addressed.

*W2: Need for high-dimensions.* The process in Alg 1, that makes pairwise distances arbitrarily similar, requires embedding dimension $n-1$ in general. This is somewhat glossed over in footnote 1. In real-world settings, this is typically not the case. At least the intrinsic dimensionality of a dataset is typically much lower than $n-1$. Moreover, many pipelines (e.g. popular visualization pipelines for single-cell data) explicitly have a PCA step that reduces dimensionality to tens of dimensions, much lower than the dataset size, before applying t-SNE. The authors should at least comment on this. Ideally, the central statements like Thm 3, 5 could make the achievable $\varepsilon$ dependent on a given dimensionality $2 \ll d \ll n-1$.

*W3: Discussion of poison points.* I much appreciate the finding that poison points exist for t-SNE. But it think its current treatment is suboptimal. The bulk of Sec 4 is on $\varepsilon$-distance perturbations, but at the end the conceptually different perturbation of adding a poison point is introduced without even starting a new paragraph or a subsection. I also think the argument on ranges of distances and "effective" distances could benefit a lot from a graphical depiction of the respective (effective) distances ranges with(out) poison point.

When reproducing the experiment using the provided code, I noticed very brittle behavior with respect to the random seed. Sometimes the embedding was as in Fig 3 with the two clusters falling into two half-discs. Sometimes it had the same shape, but the clusters were well-mixed in the disc. Other times, there was a well-mixed disc and the poison point was a very far outlier. It would be good to at least mention this brittle behavior, or even show it in additional panels.

In contrast to much of the rest of the paper, the explanation for t-SNE's behavior in the presence of a poison point remains rather vague (see also question 2). I wonder if not an integral part of that explanation should be that the poison point will act as a hub in the kNN graph (since every other point has it as its nearest neighbor). I expect that it is this affinity of the entire dataset to a common point that prevents any visual separation. However, given the random seeds for which the poison point is not embedded in the convex hull of the rest of the embedding, this is not the full story either.

Finally, the paragraph speaks of distances, not squared distances, in which case the distance to the poison point should be $1/\sqrt{2} \cdot (1-\varepsilon)$.

*W4: Proof of Lem 19:* I wonder if the proof of Lemma 19 is correct, see question 3.


**Minor**

*W5: Outlier section.* I think the novelty of section 5 is lower than that of section 4. The fact that t-SNE pulls outliers onto the main embedding is not new, e.g. [a]. While Thm 8 is new and interesting, its practical usefulness is limited, I think, as a point with separation of three times the embedding diameter clearly looks visually like an outlier. (In practice, outliers in t-SNE are of course much closer). I also wonder how the bad situation in Fig 5 right panel really is. The outliers are clearly much sparser than the dense clusters. Detecting them, e.g., with HDBSCAN on the 2D embedding should be straightforward.

*W6: Analysis of additive invariance.* Contrary to footnote three, additive invariance was extensively discussed (as "shift-invariance") by Lee and Verleysen [b] in the context of SNE.

*W7: Verification of visual hypotheses:* Since the authors mention that t-SNE is routinely used for hypothesis generation, it might make sense to also mention that it is common knowledge that these visually generated hypothesis need to be independently verified (e.g. by running a clustering method on the high-dimensional data) [c].

*W8: Presentation of proofs:* The proof of Lem 19 can be improved by mentioning that $v$ is a normalized normal vector of the hyperplane as in Definition 7 (this becomes clear later in the proof, but Def 7 does not speak of $v$ or normal vectors). Moreover, in the proof of Lemma 6, there should also be a case distinction on $D_{\text{max}}= D_{\text{min}}$ as the last two lines of the inequality becomes undefined if the embedding is a regular triangle.

*W9: Abstract* The abstract is extremely short. When reframing the storyline, this could be expanded.

**Typos:**

T1 $x_0 \in X$ not $x_0 \in\mathbb{R}^D$ in Def 7

T2: Upper summation limits in proof of Lem 18 should be $n-1$ not $n$.

T3: There are two identical equations in the proof of Thm 8. Moreover, the gradient of the loss misses a factor -4.

**References**

[a] Schubert, E., & Gertz, M. (2017). Intrinsic T-stochastic neighbor embedding for visualization and outlier detection: A remedy against the curse of dimensionality?. In International Conference on Similarity Search and Applications

[b] Lee, J. A., & Verleysen, M. (2011). Shift-invariant similarities circumvent distance concentration in stochastic neighbor embedding and variants. Procedia Computer Science, 4, 538-547.

[c] de Bodt, C., Diaz-Papkovich, A., Bleher, M., Bunte, K., Coupette, C., Damrich, S., ... & Kobak, D. (2025). Low-dimensional embeddings of high-dimensional data. arXiv preprint arXiv:2508.15929.

**Questions:**

Q1: Poison points drastically change the t-SNE output, but t-SNE's response to outliers is "muted" (line 458). But are poison points, especially the mean point in Figure 3, not also outlier points? When does the t-SNE embedding change and when is it barely affected?

Q2: How does the explanation in line 339 to 344 explain the embedding change? (a) the absolute "gap" between intra- and inter-distances remains the same ($2\varepsilon$), the ratio of gap to intra-distances is smaller, though. (b) Granted that "effective" intra- and inter-distances are more similar in some sense with the poison point, I do not see why this has to result in such a striking change in the embedding.

Q3: Should the term $2u\cdot t$ not be added rather than subtracted in the denominator in line 955? And if so, why can this positive term be dropped when constructing the lower bound in the next step?

---

> ### Author Response · Authors · 2025-11-18
>
> **W1 (presentation):**
> The reviewer is correct that there are multiple perspectives to understand the behaviors of t-SNE presented in this paper. Indeed, as the reviewer points out, the nearest-neighbor rank order is unchanged under “additive blow-up,” and therefore any rank-order based methodology will produce the same outputs.
>
> Our focus is the distance perspective, since 2D t-SNE plots implicitly convey distance information: points mapped closer together are perceived to be close to each other while those that are mapped farther away are somehow distinct. Our message from this distanced-based perspective is to emphasize that the level of (distance-based) cluster strength as indicated by a t-SNE plot is (provably) not indicative of the cluster strength of the input.
>
> This does not preclude us from not discussing these other perspectives. We fully agree with the reviewer that this distinction should be made clear throughout the paper. See below
>
> Regarding specific parts of the paper, the following will be updated:
> - Table 1, Figure 1: Clear acknowledgment that rank order of neighbors is maintained in the impostor dataset. In Figure 1, add a column showing the (single-linkage) dendograms, clearly specifying rank order being maintained while the distance-based cluster signal is weakened.
> - Intro and Section 4: A dedicated paragraph discussing that there are multiple ways of defining cluster structure, including nearest-neighbor based methods. In addition, fully discuss the various use-cases of these approaches.
> - Figure 2: Acknowledgment that these perturbations of the epsilon simplex, while maintaining distance proximity, distorts rank order.
> - Conclusion: Expanding on this discussion.
>
> **W2 (artificiality of impostor dataset):** We will emphasize in Theorems 3, 5 (and elsewhere as appropriate) that the impostor construction is not necessarily indicative of typical real-world inputs as processed by t-SNE. Additional experiments (not currently included in the paper) have indicated that even with PCA pre-processing (as done in many practical implementations of t-SNE) t-SNE has similar failure modes. We will make sure to include these results in the appendix.
>
> **W3 (poison points):** The poison point discussion will be given its own subsection, and the discussion will be expanded. In particular, we will add experiments demonstrating the robustness of the attack, e.g. different random seeds, more poison points, larger datasets, etc. The reviewer’s observation about how poison points disrupt neighborhood connectivity is precisely how we developed the idea for the attack. We will make this explicit.
>
> **W4:** Thank you for pointing out this typo; please see our answer to Q3.
>
> **W5:** The purpose of Figure 5 is to contrast t-SNE’s response to outliers vs. poison points. The figure indicates t-SNE’s representation of the data obscures the original outliers; as the reviewer correctly points out, other outlier detection strategies should be employed.
>
> It is worth noting that Theorem 8 is the first time any theoretical statement has been made regarding outliers in locally optimal t-SNE embedding. The apparent looseness of the bound in Theorem 8 stems from the fact that it can be applied to any configuration of input data points, outlier or otherwise.
>
> **W6-9:** We thank the reviewer for pointing out Lee and Verleysen’s work. We will discuss this and all the other suggestions that the reviewer makes appropriately at length.
>
> **Q1:** Both poison points and (alpha-)outliers can both be viewed as “outliers”. We refer to outliers that are inside the convex hull as poison points, whereas those that are outside are referred to as (alpha-)outliers (see Definition 7). We make this distinction simply because they are experimentally and theoretically analyzed differently.
>
> **Q2:** The discussion of how the poison point attack works will be clarified more, cf. W3.
>
> **Q3:** Thank you for pointing out the typo. Qualitatively, the bound in Theorem 8 remains the same; quantitatively, a factor of $\sqrt{2}$ needs to be included.
>
> **T1-T3:** Thank you for pointing these out.

---

> > ### Comment · Reviewer_WD8K · 2025-11-25
> >
> > Many thanks for the reply! As far as I can see, the authors did not upload a revision in which they implemented their announced changes. Since my main concern is about presentation, the exact wording matters. Therefore, I strongly recommend to upload a revised version of the manuscript highlighting the changes. Below some more comments:
> >
> > **W1**: Most suggested changes sound good, but exact phrasing matters.
> >
> > I am not fully sure about the usefulness of the single-linkage dendrogram, as it is also primarily distance-based even in high dimension. If all high-dimensional distances are very similar, then the dendrogram will have all merges at that critical distance. I am doubtful whether there will be a large difference in linkage threshold between the distances at which the clusters form and at which they merge. While the order of the merges might be the same for original and embedding, I wonder if this will be visually clear. It does not convey the idea that the (cluster) structure that t-SNE produces is in fact also present in the high-dimensional data. But seeing the revised figure might convince me otherwise. Perhaps giving a kNN graph-based metric that changes little from high-to-low dimension would be better.
> >
> > It might also help to include a color bar for the distance matrices in Fig 1 and to include the imposter distances both with a distance matrix whose colorbar starts at zero (as is currently) and one where the colorbar only covers the off-diagonal entries (which would look identical to the original data, I think).
> >
> >
> > **W2:** Very interesting that the results also hold for "typical" t-SNE visualization pipelines with PCA as intermediate step. I am curious to see these results in the revision.
> >
> > **W3-W9:** Sound good.
> >
> > **Q1:** Do you see a way to categorize outliers as poison / (alpha-)outliers before computing the embedding? I.e. based the affinity matrix?
> >
> > **Q3:** Great that the proof can be fixed so easily!

---

> > > ### Author Response · Authors · 2025-12-03
> > >
> > > W1-2: already addressed in the newly upload manuscript
> > >
> > > Q1: We are not aware of any standard/established methodology to detect poison points based solely on the affinity matrix. One heuristic may be is to subsample rows/cols of the matrix and study how the (renormalized) affinity matrix may change.

---

### Official Review · Reviewer_9aD7 · 2025-10-30

**Soundness:** 2
**Presentation:** 2
**Contribution:** 2
**Rating:** 4
**Confidence:** 3

**Summary:**

The paper analyzes the t-SNE algorithm in scenarios in which the output does not preserves the clusters within the data. Specifically, the authors consider the Silhouette score as a way to quantify the clustering property of the dataset. They then show multiple scenarios in which this property is not preserved during the embedding process, which may lead to incorrect interpretation of the data.

**Strengths:**

The paper shows interesting theoretical results relating to the pitfalls of t-SNE, when applied without applying any preprocessing beforehand.

**Weaknesses:**

1. The example given at the beginning is not clear. It is hard to understand the structure of the data, and the distance matrix of the original data is non-informative with respect to the data's structure.

2. The authors do not discuss the pitfalls of the Silhouette score (and the other scores mentioned in the Appendix). For example, given a dataset composed of two curvy lines, where each of these lines is curled around itself, the Silhouette score will be high. However, a possible embedding that preserves all the structural and cluster information can be of the form of two nearby straight lines around the origin, which may result in small Silhouette score. I am not sure that this embedding is non-informative.

**Questions:**

I wanted to emphasize again that the results are interesting, but I do have some concerns. I will be happy to update my score based on your responses.

1. The authors examine a framework in which the number of samples n and dimension d satisfy d=n-1.
1A. Could the authors elaborate more on why this framework is interesting? Isn't this regime too restrictive?
1B. What was the framework that other works used to tackle related questions with respect to t-SNE?
1C. A common practice in the field is to apply PCA on the data before applying t-SNE. Are your results transferable to such a case?
The paper [Kobak et al], which you cited in your work, states in its abstract: "(we) develop a protocol for creating more faithful
t-SNE visualizations .... it includes PCA initialization".
Additionally, the standard preprocessing for single-cell PBMC data, such as the one shown in Figure 1, includes PCA.

2. In Theorem 5, I am missing a relation between epsilon and the norm of the points within X and X'. If the norm of the points is much smaller than epsilon, the implications of this theorem may be weak.

3. Your finding about the "Additive invariance with respect to the pairwise squared distance" requires clarification. Throughout the paper the authors indicate that they have identified this property, but while this is a straightforward property of normalized kernel, there are multiple works that uses this fact in their proofs already.
For example, there are papers that study effect of homoskedastic noise on kernel matrices (See [El Karoui]) and normalized Laplacians (See [El Karoui et Al]) in high-dimensional settings. While the first paper shows that that the noise results in global multiplicative factor to the kernel matrix (much like adding a global constant to the pairwise distances), the second shows how the normalization helps removing or reducing the noise, and uses a similar observation throughout its proof to the one made in this paper.
Furthermore, this was noted in Section 1.3 "The influence of noise", of [Landa et Al.], using a similar description to the one used in the current paper, via a global constant that is added to the squared pairwise Euclidean distances and removed through the normalization.


Minor adjustments:
A. Figures captions. The figure in page 1 and the figure on the top of page 2 are missing the captions. I am not sure whether this is valid.
B. It will be good to refer to Appendix A in Table A (page 1) as you did in the second paragraph of Section 4 (page 4).


Reference:
Dmitry Kobak, and Philipp Berens. The art of using t-SNE for single-cell transcriptomics. Nature Communications, 2019.
Noureddine El Karoui. n information plus noise kernel random matrices. The Annals of Statistics, 2010.
Noureddine El Karoui, and Hau-Tieng Wu. "Graph connection Laplacian methods can be made robust to noise. The Annals of Statistics, 2016.
Boris Landa, and Xiuyuan Cheng. Robust inference of manifold density and geometry by doubly stochastic scaling. SIAM Journal on Mathematics of Data Science, 2023.

---

> ### Author Response · Authors · 2025-11-18
>
> **W1/W2:** The reviewer is correct that inspecting just the raw distance matrix of the original data may not fully communicate the data’s intrinsic structure. We are specifically trying to understand cluster structure in the input data and its visualization; block-diagonal decompositions of the distance matrices are often used for this purpose (e.g. spectral clustering, subspace clustering, etc). The purpose of presenting the distance matrix is to emphasize that there is a strong qualitative difference between the t-SNE plot’s block cluster structure in contrast to that of the original dataset.
>
> This perspective directly leads us to use the silhouette score for measuring clustering strength, as it closely relates to the strength of block diagonal structure in the distance matrix. By the same token, as the reviewer observes, it may not be the right tool for analyzing non-spherical clusters.
>
> In light of these points and the points raised by other reviewers (see W1 of Reviewer WD8K), we will expand our discussion of Intro/Table 1/Figure 1 to include neighbor-based rank-order clustering (e.g. hierarchical clustering) and non-spherical clusters for a more balanced treatment. In particular, we will explain the shortcomings of widely-used cluster analysis techniques like silhouette and block-diagonal decompositions.
>
> **Q1**
> - **A/B/C:** We understand how, when the input dimension D = n-1, the result may seem restrictive and perhaps even uninteresting. However, the assumption D = n-1 is made without loss of generality. When D > n-1, one can always project the data down to the (n-1)-dimensional affine space spanned by the n points, thus bringing the data dimension down to n-1 without changing any distance structure. When D < n-1, one can simply pad the D-dimensional data vectors with zeros to bring the data dimension to n-1. We will make this clear.
> - **B:** While previous works don’t explicitly make this D=n-1 assumption, they operate under the same level of generality regarding the data dimension.
> - **C:** Please see point W2 in our response to Reviewer WD8K.
>
> **Q2:** The reviewer’s concern is relevant for additive notions of error. Theorem 5 is a statement about multiplicative error: since the middle term in lines 290-291 is a ratio between distances, the statement remains meaningful regardless of the absolute magnitude of interpoint distances. For example, consider eps = 0.1 with the data diameter of X and X’ being 0.01 (i.e. 1/10 of eps). Theorem 5 asserts that the ratios of respective interpoint distances between X and X’ are within 1+eps (i.e. 10%) of each other.
>
> **Q3:** We thank the reviewer for bringing these works to our attention. Our work focuses on the relevance of additive invariance with respect to the false positive behavior of t-SNE and the potential pitfalls of doing data analysis on t-SNE plots. We will be sure to emphasize that we are not the first to discover this property and that the other works have used this property of normalized kernels in their analyses.

---

### Official Review · Reviewer_wvWe · 2025-10-31

**Soundness:** 3
**Presentation:** 3
**Contribution:** 3
**Rating:** 6
**Confidence:** 3

**Summary:**

The paper tackles an important but often overlooked issue with t-SNE, that is, whether the visual clusters we see actually reflect any true structure in the data. The authors provide both theoretical and empirical results showing that t-SNE can produce identical, strongly clustered embeddings even from unclustered data, and that it fails to represent extreme outliers. The analysis attributes these effects to two intrinsic properties: additive invariance of squared distances and asymmetry between input and output affinity matrices.

**Strengths:**

- The focus on false positives in cluster visualization is fresh and genuinely useful.
- The theoretical results (especially Theorems 3, 5, and 8) are carefully stated and, to my reading, technically sound. They connect interestingly with known properties of t-SNE optimization, like the concentration of measure effect in high dimensions.
- The idea of constructing “impostor” datasets that generate the same clustered t-SNE plots is a good demonstration.

**Weaknesses:**

- In the paper, the empirical results are shown in $d=2$. This is understandable for visualization, but it limits the evidence for the general claims. It remains unclear whether the same cluster exaggeration or outlier suppression occurs when t-SNE embeds into 3D or slightly higher dimensions (where some of the “crowding” artifacts might diminish). The additive-invariance argument may hold mathematically, but its practical severity could depend on $d$. The author could think of a way to quantify the severity, e.g., by visualization (for 3D) or KNN accuracy measure.

- The same figure contrasts t-SNE’s collapse with PCA’s stability, but PCA’s robustness is expected because it is linear and insensitive to single points at the mean. In contrast, t-SNE minimizes KL divergence on a normalized graph, so central points have a disproportionate effect by design. The figure thus compares incomparable objectives. A fairer baseline would be UMAP or TriMap, which share t-SNE’s nonlinear affinity structure.

**Questions:**

Lemma 15 is missing in the current manuscript.

Does a single poison point still break the structure for $n=2000$ or $n=10000$? Otherwise, the example in Figure 3 might simply reflect finite-sample instability rather than a general property.

---

> ### Author Response · Authors · 2025-11-18
>
> **W1 (output dimension):** The results of the paper (namely Theorems 3,5,8) are valid for any fixed output dimension. The reviewer brings up an excellent point to verify this in practical settings. Additional experiments (not currently included in the paper) suggest that the empirical observations in the paper (e.g. impostor datasets, merging of outliers and clusters, poison points, etc) persist in d=3. We will include these experiments in the appendix.
>
> **W2**
> - **(choice of comparing with PCA):** The reviewer is correct to point out that PCA and t-SNE are quite different. PCA is chosen as the “foil”/baseline/control for comparison purposes because it is arguably the simplest and most well-understood dimension reduction technique. One would expect that if something as simple as PCA can reveal interesting structure, then more sophisticated techniques like t-SNE do so as well.
> - **(comparison with UMAP, etc.):** Empirically, experiments suggest that the failure modes demonstrated on t-SNE extend to other force-based dimension reduction techniques like UMAP and TriMap (see also response to Reviewer 1Dsn, W2). We will include these observations in the appendix.
>
> **Q1**: Lemma 15 is in the supplementary material, see line 724.
>
> **Q2 (poison point for larger n):** We have run this experiment for large n, up to 10,000, and the results are consistent with Figure 3. We will include the large scale experiments in the Appendix. See also our response to Reviewer WD8K, point W3.

---

### Official Review · Reviewer_1Dsn · 2025-11-02

**Soundness:** 3
**Presentation:** 3
**Contribution:** 2
**Rating:** 4
**Confidence:** 3

**Summary:**

This paper introduces a novel theoretical framework for understanding how t-distributed Stochastic Neighbor Embedding (t-SNE) can fabricate or distort clusters and outliers—a phenomenon that has been largely underexplored. The authors substantiate their claims with rigorous theoretical analysis and extensive empirical studies.

**Strengths:**

1. This paper addresses an interesting problem that t-SNE can misrepresent the cluster structure and ignore outliers of the original data, which is relatively underexplored before.
2. The authors provide rigorous theoretical analysis and extensive empirical studies to support their claims.

**Weaknesses:**

1. **Limited scope:** The study focuses only on t-SNE’s misrepresentation of cluster structures and the treatment of outliers. However, the authors do not propose potential solutions to mitigate these challenges or develop a theoretical framework that could be extended to broader dimension reduction techniques or visualization methods.
2. **Limited theoretical novelty:** The theoretical analysis mainly relies on the exponential operations used in t-SNE, involving only relatively simple algebraic calculations. This results in limited analytical depth and weak theoretical contribution, which may hinder the development of a deeper understanding of the observed empirical phenomena.

**Questions:**

1. Could the authors provide more intuitive interpretation of additive invariance?
2. Is the theoretical analysis applicable to variants of t-SNE, such as FFT-t-SNE?
3. Since the misrepresentation of cluster structure does not occur for PCA in the comparison experiments, could the authors provide further theoretical explanation for this phenomenon?
4. There are other popular dimensionality reduction techniques, such as UMAP. Can the proposed theoretical framework be extended to cover these methods as well?
5. In the preliminaries section, the authors assume that the input dimension $D = n - 1$. However, in high-dimensional statistics, it is more common to consider $D \gg n$. In practical applications such as few-shot learning in image processing, the input dimension is often much larger than the sample size. Could the theoretical analysis be extended to cover these scenarios?

---

> ### Author Response · Authors · 2025-11-18
>
> **W1 (scope):**
> - **(the specific choice of using t-SNE)** We chose to focus on t-SNE because it is generally representative of force-based dimension reduction procedures; it is one of the most widely used; and it is by far the most well-studied on the theoretical front.
> - **(developing an extended theoretical framework)** In light of existing work (see e.g. the paper cited on line 122), there are known general failure modes for all data visualization methods. Our paper focuses on t-SNE-specific failure modes. Preliminary experiments suggest that they extend to UMAP and other force-based dimension reduction techniques.
>
> **W2 (theoretical simplicity and novelty):** We appreciate that the reviewer recognizes the effort we put into presenting our proofs and techniques in a simple and intuitive manner on a subject that has largely resisted theoretical study for over a decade (as also recognized by Reviewer WD8K, S4).
>
> **Q1**
> - **(what is additive invariance?):** When a function of data, like t-SNE, is "additively invariant," we should think of it as only "registering" the relative (additive) differences between pairwise distances – the magnitude of those distances are irrelevant.
> - **(why intuitively t-SNE is additive invariant?)** To work in practical data regimes, t-SNE needs to be noise-tolerant. This means that even when the clusters in data are "muddled" by noise, it should be able to tease out the underlying structure. Additive invariance property enables t-SNE to extract salient cluster structure (e.g. Figure 1). See (Lee and Verleysen 2011; Shift-invariant similarities circumvent distance concentration in stochastic neighbor embedding and variants).
>
> We will clarify these intuitions in the final version.
>
> **Q2:** The results of Section 4 applies to any t-SNE variant that has the additive invariance property. The results of Section 5 are specific to the t-SNE loss function but would still apply to something like FIt-SNE since it is an approximation to the same objective.
>
> **Q3:** The ability of PCA to preserve cluster structure on well-clustered input data (e.g. data coming from a mixture of well-separated log-concave distributions) has been studied extensively in prior theoretical computer science work, see, e.g. [Vempala and Wang 2004: A spectral algorithm for learning mixture models]. We will add a discussion on this in the related works section.
>
> **Q4:** As discussed in response to W1: empirically, UMAP and other force-based data visualization methods have similar failure modes as t-SNE. On the theory front, establishing these connections rigorously is a line of continuing work.
>
> **Q5:** Please see our response to Reviewer 9aD7’s Q1A.

---

### Author Response · Authors · 2025-11-18

We thank the reviewers for their detailed and insightful comments which will surely improve this paper.

---

### Author Response · Authors · 2025-12-03

An updated version of the manuscript that addresses the key points that were raised by the reviewers has been uploaded for their consideration. We thank all the reviewers for their hard work and helping us improve our work.

---

### Meta-Review · Area_Chair_1dHR · 2025-12-17

**Summary:**

The reviewers provided a few constructive suggestions and have the following concerns: 1) The presentation should be enhanced; 2) Some explanations for mathematical formulas and theoretical claims are insufficient; 3) The assumption of $n=D-1$  is too restrictive; 4) More evaluation metrics should be used.

After reading the paper thoroughly, I believe that these concerns are either addressed or minor issues.

The major concern that cannot be addressed is from Reviewer cJrt: The paper doesn't provide concrete guidance on how users should interpret or adjust their use of t-SNE in light of the findings in the paper. The reviewer maintains the negative assessment after the rebuttal.

I have an additional comment. In all experiments, the authors didn't show the specific values of the hyperparameter 'perplexity'. I read the code briefly and found that sometimes the perplexity is fixed to 20, and sometimes it is the default value (e.g. 30). According to my experience, a lower perplexity often leads to a looser visualization and more isolated points or small clusters. It is not clear to the readers whether the intuitive findings given by the experiments remain unchanged or not, although the theoretical analysis applies to any perplexity value. I hope that the authors could consider these issues and add more experiments and discussion in the new version of the paper.

Given the interesting numerical findings and supportive theoretical analysis in the paper, I recommend acceptance.

**Reviewer Concerns:**

**Reviewer 1Dsn**
* The study focuses only on t-SNE’s.  It could be extended to other DR techniques. (I think this is a minor issue)
* The paper has limited theoretical novelty.
* The theoretical analysis relies on the assumption that the number of data points is greater than the feature dimension. (I think this is a minor issue)

**Reviewer wvWe**
* Instead of 2D visualization and Silhouette metric, the authors may consider 3D visualization and other metrics (e.g., KNN accuracy). (The authors haven't addressed this concern)

**Reviewer 9aD7**
* The example given at the beginning hasn't been clearly explained.
* The authors do not discuss the pitfalls of the Silhouette score (and the other scores mentioned in the Appendix).

After reading the rebuttal and the paper, I think the two concerns have been addressed by the response and revision.

**Reviewer WD8K**
* The presentation should be improved. (This has been or can be addressed easily)
* The qualitative and quantitative evaluation should be improved. For instance, other popular clustering metrics, like the modularity of the kNN graph, its conductance, or its spectral gap, etc.
* The paper didn't analyze the impact of data preprocessing (e.g. PCA) on the performance of tSNE.
* The discussion of poison points is insufficient.
* One concern regarding the correctness of the proof for Lemma 19.

Most of these issues have been addressed by the rebuttal.

**Reviewer cJrt**
* The paper does not examine whether the reported phenomena (e.g., exaggeration or misrepresentation) persist under early-stopping conditions in the optimization of tSNE.
* The paper doesn't provide concrete guidance on how users should interpret or adjust their use of t-SNE in light of the findings in the paper.

After the rebuttal, the reviewer stated that the overall assessment of the paper remained unchanged.

**Reviewer Scores:**

Reviewer 1Dsn may raise the score to 6, Reviewer wvWe may keep the score 6 unchanged, Reviewer 9aD7 may raise the score to 6, and Reviewer cJrt will keep the score 4 unchanged.

---

### Decision · Program_Chairs · 2026-01-26

Accept (Poster)